# Temporal Causal Mechanism Transfer for Few-shot Action Recognition

## Abstract

The goal of few-shot action recognition is to recognize actions in video sequences for which there exist only a few training samples. The challenge is to adapt a base model effectively and efficiently when the base and novel data have significant distributional disparities. To this end, we learn a model of a temporal causal mechanism from the base data by variational inference. When adapting the model by training on the novel data set we hold certain aspects of the causal mechanism fixed, updating only auxiliary variables and a classifier. During this adaptation phase, we treat as invariant the time-delayed causal relations between latent causal variables and the mixing function that maps causal variables to action representations. Our experimental evaluations across standard action recognition datasets validate our hypothesis that our proposed method of Temporal Causal Mechanism Transfer (TCMT) enables efficient few-shot action recognition in video sequences with notable performance improvements over leading benchmarks.

## 1 Introduction

Supervised action recognition continues to be an active and productive area of research (Xing et al., 2023b; Ahn et al., 2023; Zhou et al., 2023; Zhang et al., 2023). However, given the immense variety of possible actions in the real world, there is a natural challenge of learning action representations with little training data. One promising approach for overcoming this lack of labeled data is few-shot learning (Cao et al., 2020; Perrett et al., 2021; Thatipelli et al., 2022). Few-shot learning involves training a model on a large (base) dataset, and then using a small number of samples from another (novel) dataset to update the model (e.g., tune parameters). We apply few-shot learning to action recognition, where the action labels in the novel data differ from the base data, and these datasets can be drawn from significantly different distributions.

One popular approach to few-shot action recognition is to tune a feature extractor and/or classifier, as done by ORViT (Herzig et al., 2022), SViT (Ben Avraham et al., 2022) and ActionCLIP (Wang et al., 2021). Few-shot action recognition can be made more efficient by fixing parts of the learned model that do not get updated during the adaptation phase, as long as this can be done without sacrificing model performance. For example, VideoPrompt (Ju et al., 2022) tunes a feature extractor only partially and VL Prompting (Rasheed et al., 2023) only tunes an ancillary module on the novel data. Several metric-based approaches have been proposed in which a metric space is learned from base data and assumed to transfer to novel data without adjustment (Wang et al., 2023c; Xing et al., 2023a). Generally, a major challenge is to determine which aspects of the learned model should be held fixed and which should be updated for efficient and effective adaptation, which depends on the distributional disparities between the base and novel data.

The causes and nature of distributional disparities are largely uninvestigated in the current literature on few-shot action recognition. If we can isolate the factors that cause the distributional disparities, we can update only that part of our model during adaptation. To this end, our approach builds on recent advances in causal representation learning (Schölkopf et al., 2021; Huang et al., 2022; Feng et al., 2022; Kong et al., 2022; Xie et al., 2023). We learn a generative model which consists of a transition function that models the transitions of latent causal variables and a mixing function that models how the latent variables determine observable action representations over time. At each time instant the causal variables evolve according to a transition function, and the mixing function generates the action representation accordingly. However, formal descriptions of the factors that

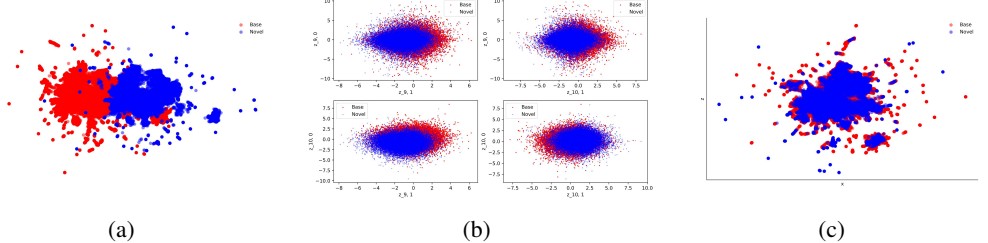

|     |     |     |
| --- | --- | --- |
| (a) | (b) | (c) |

Figure 1: Motivating example of Temporal Causal Mechanism Transfer (TCMT) for few-shot action recognition (*The original PDF files for this Figure are available in zip of supplementary, providing additional and clearer illustration*) : (a) shows the UMap visualization of the action feature embeddings obtained by a fixed ViT-B/16 backbone (Radford et al., 2021) on the base data (red) and novel data (blue) in the Sth-Else dataset. The obvious distribution disparities demonstrate the difficulty of few-shot learning for action recognition; (b) shows an example of pairplot of the values of latent causal variables from two trained models, one trained only on the base data (red) and the other only on the novel data (blue) from Sth-Else. (c) shows a pairplot of UMap projections for action feature embeddings $x$ and latent variables $z$ from two models: one trained on base data (red) and the other on novel data (blue) from the Sth-Else dataset. The near-perfect alignment in (b) and (c) supports our hypothesis of invariant transition and mixing functions across base and novel data.

cause the disparities cannot be done without further assumptions. Our central assumption is that the base data and novel data share certain aspects of the temporal causal mechanism – namely, transition function and mixing function – and that an auxiliary variable captures the disparate aspects of the two data distributions. We therefore propose a Temporal Causal Mechanism Transfer (TCMT) approach, in which we assume that the distribution discrepancies between the base and novel data stem from an auxiliary variable, allowing us to update this auxiliary variable along with the action classifier during adaptation while holding the transition function and mixing function fixed.

Figure 2a illustrates an example of the generative model we use to describe the temporal causal process. The time-delayed causal relations of the $N$-dimensional latent causal variables ($\mathbf{z}$) are represented by the arrows going left-to-right. The transition from $\mathbf{z}_{n,t}$ ($n \in [1, N]$) from $t = 1$ to $t = 2$ is influenced by the auxiliary context variable $\boldsymbol{\theta}_t$. At time $t$, the action representations $\mathbf{x}_t$ are generated by $\mathbf{z}_t$ as represented by the downward arrows at the bottom. The sequence-level action class $\mathbf{y}$ is predicted from $\mathbf{z}$. Figure 2b demonstrates that modeling distribution disparities using causal representation can lead to a more efficient and effective adaptation to novel data.

Figure 1 showcases a motivating example of a few-shot action recognition task on the Sth-Else dataset (Materzynska et al., 2020) with TCMT. The Sth-Else dataset is marked by large distribution disparities between the base and novel data, as seen in Figure 1a, making few-shot action recognition task challenging. We learned two causal mechanisms for comparisons — one from the base data and the other from the novel data. As indicated in Figure 1b, the near-perfect alignment was found between the transitions of $\mathbf{z}$ from the two causal mechanisms, thus validating our hypothesis that the transition function can remain invariant. Similarly, Figure 1c validates our assumption that the mixing function can remain fixed, as evidenced by the overlap of the learned mixing functions on the base and novel data. Lastly, Figure 2b highlights the efficiency and effectiveness of TCMT at handling the distributional disparities on Sth-Else.

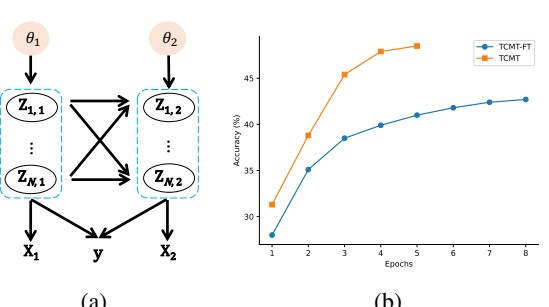

|     |     |
| --- | --- |
| (a) | (b) |

Figure 2: (a) The temporal causal process underlying the data generation for $t \in \{1, 2\}$. (b) The top-1 accuracy obtained by our proposal versus a baseline model (TCMT-FT) in which all parts of the model are updated on the novel data, demonstrating that TCMT is able to achieve better outcomes with fewer training epochs.

**Contributions.** **(1)** We improve on existing methods for few-shot action recognition by using a generative model based on causal representation learning. **(2)** Our approach, Temporal Causal Mechanism Transfer (TCMT), centers on learning a temporal causal mechanism. The mechanism's transition function that governs changes to the latent causal variables and it's mixing function that

generates action representations do not need to be updated during adaptation, so that only the auxiliary context variable that captures the distribution shifts between the base and novel data and the classifier need to be updated. **(3)** We validate TCMT through experimental results and comprehensive ablation studies, showing that TCMT can achieve accuracy comparable to or better than leading benchmarks with less parameter updating.

## 2 METHODOLOGY

A typical few-shot setting has two disjoint sets of data; base and novel. Let $\mathcal{D} = (\mathbf{v}_i, \mathbf{y}_i)_{i=1}^{I}$ denote the base data used for initial training, where $\mathbf{v}_i$ is the $i$-th video sequence and $\mathbf{y}_i \in \mathcal{C}_{base}$ is the corresponding action label in the class of base labels. The novel data is comprised of two parts, the support set used for updating the model $\mathcal{S} = (\mathbf{v}_j, \mathbf{y}_j)_{j=1}^{J}$ where $\mathbf{y}_j \in \mathcal{C}_{novel}$, and the query set used for inference $\mathcal{Q} = (\mathbf{v}_{j'}, \mathbf{y}_{j'})_{j'=1}^{J'}$ where $\mathbf{y}_{j'} \in \mathcal{C}_{novel}$. Notably, there only exist limited samples for $\mathcal{S}$, e.g., $J = 5$, and $\mathcal{C}_{base} \cap \mathcal{C}_{novel} = \emptyset$. The goal of few-shot action recognition is to correctly identify $\mathbf{y}_{j'}$ for the samples in $\mathcal{Q}$ using a model initially trained on $\mathcal{D}$ and further updated on $\mathcal{S}$.

### 2.1 GENERATIVE MODEL

The generative model of our temporal causal process, as shown in Fig. 2, is assumed the action representations and action labels are each generated from a set of latent causal variables $\mathbf{z}$. Let $\mathbf{z} = \{\mathbf{z}_t\}_{t=1}^{T}$ denote the latent causal variables at time steps $t = 1 \ldots T$ and let $\mathbf{x} = \{\mathbf{x}_t\}_{t=1}^{T}$ denote the action representations. We assume that the action representations are generated by a non-linear, invertible **mixing function** over the latent variables:

$$\mathbf{x}_t = g(\mathbf{z}_t) \tag{1}$$

and the action label predictions are causally determined by the latent variables via a **classifier**:

$$\mathbf{y} = e(\mathbf{z}) \tag{2}$$

In our model there are $N$ causal variables. For $n \leq N$ the value of latent variable $\mathbf{z}_{n,t} \in \mathbf{z}_t$ at time $t$ is determined by the values of its time-delayed parents $\mathrm{Pa}(\mathbf{z}_{n,t}) \subseteq \mathbf{z}_{t-1}$ at the previous time step (i.e., the set of latent factors that directly cause $\mathbf{z}_{n,t}$) as well as auxiliary variable $\boldsymbol{\theta}_t$ and the noise term $\epsilon_{n,t}$:

$$\mathbf{z}_{n,t} = f_n(\mathrm{Pa}(\mathbf{z}_{n,t}), \boldsymbol{\theta}_t, \epsilon_{n,t}) \tag{3}$$

Here $f$ is referred to as the **transition function** and is assumed to be invertible. Note that $\mathrm{Pa}(\mathbf{z}_{n,t})$, $\boldsymbol{\theta}_t$, and $\epsilon_{n,t}$ are mutually independent, and we assume that $\mathbf{z}_{n,t}$ and $\mathbf{z}_{n',t}$ are conditionally independent conditioned on $\mathbf{z}_{t-1}$ for all such pairs of latent causal variables.

Looking ahead, during the adaptation phase for learning on the support set $\mathcal{S}$, we learn a new sequence of the auxiliary context variables $\boldsymbol{\theta}$ and updated weights for classifier $e$ while the mixing function $g$ and transition function $f$ will be held fixed (not updated). This reflects our hypothesis that certain aspects of the temporal causal process will be invariant across the base and novel datasets.

### 2.2 NETWORK

Given base data $\mathcal{D}$ and support data $\mathcal{S}$ with action representations $\mathbf{x}$, we can learn a model for action recognition based on the temporal causal process by extending the framework of Conditional Variational Auto-Encoders (CVAE) (Sohn et al., 2015). See Table 14 in the appendix for details on our network architecture.

**Temporal Prior Estimation by Prior Network.** First, observe that we can express the probability distribution of $\mathbf{z}_t$ in terms of conditional transition priors $p(\mathbf{z}_{n,t}|\mathbf{z}_{n,t-1}, \boldsymbol{\theta}_t)$. However, we do not have an explicit form for $p(\mathbf{z}_t|\mathbf{z}_{t-1}, \boldsymbol{\theta}_t)$ and it must be learned. Since the transition function $f$ is invertible, we can estimate the inverse transition function such that the distribution of $\hat{f}_n^{-1}(\hat{\mathbf{z}}_{n,t}, \mathrm{Pa}(\hat{\mathbf{z}}_{n,t}), \boldsymbol{\theta}_t)$ matches an assumed noise distribution $p(\epsilon_{n,t}|\boldsymbol{\theta}_t)$. We can then use the inverse transition function to estimate the priors:

$$p(\hat{\mathbf{z}}_{n,t}|\hat{\mathbf{z}}_{n,t-1}, \boldsymbol{\theta}_t) = p\big(\hat{f}_n^{-1}(\hat{\mathbf{z}}_{n,t}, \mathrm{Pa}(\hat{\mathbf{z}}_{n,t}), \boldsymbol{\theta}_t)\big) | \frac{\partial \hat{f}_n^{-1}}{\partial \hat{\mathbf{z}}_{n,t}} | \tag{4}$$

Given that $\mathbf{z}_{n,t}$ and $\mathbf{z}_{n',t}$ are conditionally independent conditioned on $\mathbf{z}_{t-1}$ for all latent causal variables $n$ and $n'$, we can construct an estimate for $\mathbf{z}_t$ by:

$$p(\hat{\mathbf{z}}_t|\hat{\mathbf{z}}_{t-1}, \boldsymbol{\theta}_t) = \prod_{n=1}^{N} p(\hat{\epsilon}_{n,t}|\boldsymbol{\theta}_t)|\frac{\partial \hat{f}_n}{\partial \hat{\mathbf{z}}_{n,t}}^{-1}| \tag{5}$$

For a detailed derivation please refer to Appendix A.1.

**Temporal Posterior Estimation by Encoder.** We express the posterior using $q(\hat{\mathbf{z}}_t|\mathbf{x}_t)$. We assume $\mathbf{z}_t$ is conditionally independent of all $\mathbf{z}_{t'}$ for $t' \neq t$ conditioned on $\mathbf{x}$, and therefore we can decompose the joint probability distribution of the posterior by $q(\hat{\mathbf{z}}|\mathbf{x}) = \prod_{t=1}^{T} q(\hat{\mathbf{z}}_t|\mathbf{x}_t)$. We choose to approximate $q$ by an isotropic Gaussian characterized by mean $\mu_t$ and covariance $\sigma_t$. We use the encoder portion of our CVAE to learn the posterior by an MLP followed by leaky ReLU activation:

$$\hat{\mathbf{z}}_t \sim \mathcal{N}(\mu_t, \sigma_t) \text{ where } \mu_t, \sigma_t = \text{LReLU}(\text{MLP}(\mathbf{x}_t)) \tag{6}$$

**Action Representation Reconstruction by Decoder.** The decoder portion of our CVAE models the mixing function $g$ as it reconstructs an estimate of the action representations $\hat{\mathbf{x}}_t$ from the estimated latent causal variables $\hat{\mathbf{z}}_t$ sampled from the estimated posterior. We implement the decoder using a stacked MLP followed by leaky ReLU activation.

$$\hat{\mathbf{x}}_t = \text{LReLU}(\text{MLP}(\hat{\mathbf{z}}_t)) \tag{7}$$

**Classifier.** We train a classifier for each video from the sequence of latent variables. Our classifier consists of an MLP:

$$\hat{\mathbf{y}} = \text{MLP}(\text{Concat}(\hat{\mathbf{z}})) \tag{8}$$

where Concat($\hat{\mathbf{z}}$) concatenates $\mathbf{z}_1, \ldots, \mathbf{z}_T$ along the time dimension.

**Context Network for $\theta$ modeling.** Since $\boldsymbol{\theta}$ is not directly observable from the real-world videos, we use a two-layer ConvLSTM (Shi et al., 2015) that takes $\mathbf{x}$ as input to model $\boldsymbol{\theta}$. At time $t$ we define $\boldsymbol{\theta}_t$ as:

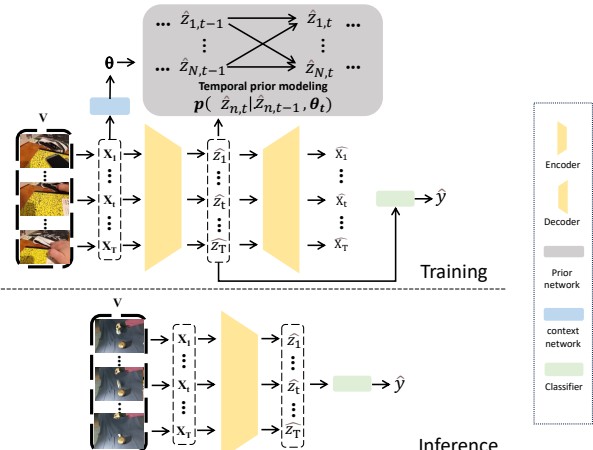

Figure 3: **Overall framework of TCMT.** The arrows indicate information flow. During the training on $\mathcal{D}$, we first learn the posterior $\hat{\mathbf{z}}$ through the encoder given the action representations $\mathbf{x}$ from a video sequence $\mathbf{v}$. The prior network then outputs an estimate of the noise model to estimate the temporal prior. The sequence of context variables $\boldsymbol{\theta}$ serves as input for the prior network along with $\hat{\mathbf{z}}$. The decoder reconstructs the action representation $\hat{\mathbf{x}}$ and the classifier predicts the action class $\hat{\mathbf{y}}$ based on the latent variables. Note that only $\boldsymbol{\theta}$ and the classifier network are updated during adaptation on $\mathcal{S}$ while the rest of the model remains fixed. During inference, we sample $\hat{\mathbf{z}}$ from the encoder and predict the action class $\hat{\mathbf{y}}$ using our classifier $e(\hat{\mathbf{z}})$.

$$\boldsymbol{\theta}_t = \text{ConvLSTM}(\mathbf{x}_t, \boldsymbol{\theta}_{t-1}) \tag{9}$$

## 2.3 LEARNING AND INFERENCE

**Loss Functions.** Our overall loss function for action recognition combines the classification loss and evidence lower bound (ELBO):

$$\mathcal{L} = \mathcal{L}_{\text{ELBO}} + \mathcal{L}_{\text{cls}} \tag{10}$$

The ELBO loss combines the reconstruction loss from using our decoder to estimate $\mathbf{x}_t$ with the KL-divergence between the temporal posterior and temporal prior.

$$\mathcal{L}_{\text{ELBO}} = \sum_{t=1}^{T} \left( \mathcal{L}_{\text{Recon}}(\mathbf{x}_t, \hat{\mathbf{x}}_t) - \beta \sum_{n=1}^{N} \mathcal{L}_{\text{KLD}} \right) \tag{11}$$

where the KL divergence is

$$\mathcal{L}_{\text{KLD}} = \mathbb{E}_{\hat{\mathbf{z}}_{n,t} \sim q} \log q\left(\hat{\mathbf{z}}_{n,t} | \mathbf{x}_t\right) - \log p(\hat{\mathbf{z}}_{n,t} | \hat{\mathbf{z}}_{n,t-1}, \boldsymbol{\theta}_t) \tag{12}$$

Here, $\mathcal{L}_{\text{Recon}}$ measures the discrepancy between $\mathbf{x}_t$ and $\hat{\mathbf{x}}_t$ using binary cross-entropy, and $\beta$ acts as a hyperparameter that balances the reconstruction loss and KL-divergence.

For the classification loss, we examine both the cross-entropy loss using a one-hot encoding of $\mathbf{y}$ as well as the contrastive loss using a text embedding of $\mathbf{y}$. With cross-entropy loss,

$$\mathcal{L}_{\text{cls}} = -\mathbb{E}_{\hat{\mathbf{y}}}\big(\text{one-hot}(\mathbf{y}) \cdot \log(\text{softmax}(\hat{\mathbf{y}}))\big) \tag{13}$$

and with NCE loss (Gutmann & Hyvärinen, 2010) with a temperature parameter $\tau$,

$$\mathcal{L}_{\text{cls}} = -\sum_i \log \frac{\exp(\text{sim}(\hat{\mathbf{y}}_i, \text{embed}(\mathbf{y}_i))/\tau)}{\sum_m \exp(\text{sim}(\hat{\mathbf{y}}_i, \text{embed}(\mathbf{y}_m))/\tau)} \tag{14}$$

where $sim(\cdot, \cdot)$ is the cosine similarity between the text embedding of the action label and our prediction for the $i^{\text{th}}$ video sequence. When we use cross-entropy for classification loss we denote our model by **TCMT**$_H$, and use **TCMT**$_C$ when using NCE loss.

**Adaptation.** After training on the base data $\mathcal{D}$, we freeze the weights of the encoder, decoder, and prior network so that the learned functions $f$ and $g$ remain fixed. When updating by training on the novel data in $\mathcal{S}$ only the weights associated with the context network and classifier network are updated.

**Inference.** To perform inference we sample $\hat{\mathbf{z}}$ according to Eq. 6 using our encoder, and we either choose the maximum value (highest probability) from the prediction $\hat{\mathbf{y}}$ or choose the label whose text embedding maximizes cosine similarity.

## 3 EXPERIMENTS

### 3.1 EXPERIMENTAL SETUP

We carry out two types of few-shot learning experiments; all-way-k-shot and 5-way-k-shot. In all-way-k-shot we try to classify all action classes in the class for the novel dataset ($\mathcal{C}_{novel}$), while in 5-way-k-shot learning we only try to estimate 5 label classes at a time in a series of trials. The number of shots (k) refers to the number of training samples in $\mathcal{S}$ available for each action label. Once we partition our data into base set $\mathcal{D}$ and novel set $\mathcal{S} \cup \mathcal{Q}$, the number $k$ determines how many samples for each action class we choose for $\mathcal{S}$ and the rest is used for the query set $\mathcal{Q}$.

**Datasets.** For the experiments that perform training and testing on the Sth-Else dataset, we use the official split of data (Materzynska et al., 2020; Herzig et al., 2022; Ben Avraham et al., 2022). Similarly, for experiments using other datasets, we split the data into $\mathcal{D}$, $\mathcal{S}$, and $\mathcal{Q}$ in accordance with the prior work we use as benchmarks, as described late. The other datasets we use for novel data are SSv2 (Goyal et al., 2017), SSv2-small (Zhu & Yang, 2018), HMDB-51 (Kuehne et al., 2011), and UCF-101 (Soomro et al., 2012).

**Performance Measure.** In all experiments we compare the Top-1 accuracy, i.e., the maximum accuracy on any action class, of TCMT against leading benchmarks for few-shot action recognition. The results of the top-performing model are given in **bold**, and second-best are underlined.

**Implementation Details.** In each experiment there is a **backbone** used to extract the action representations $\mathbf{x}_t$ used for training, and in all experiments the backbone is either ResNet-50 (He et al., 2016) or ViT-B/16 (Radford et al., 2021) trained on the base data set $\mathcal{D}$, where the choice of backbone matches what was used in the benchmark experiments. We use the AdamW optimizer Loshchilov & Hutter (2019) and cosine annealing to train our network with a learning rate initialized at 0.002 and weight decay of $10^{-2}$. For all video sequences we use $T = 8$ uniformly selected frames and to compute the ELBO loss we choose $\beta = 0.02$ to balance the reconstruction loss and KL-divergence. Also we set $\tau = 0.07$ for the NCE loss. For a detailed summary of our network architectures see Table 14 in the appendix. Our models are implemented using PyTorch, and experiments are conducted on four Nvidia GeForce 2080Ti graphics cards, supplied with 44 GB memory in total.

## 3.2 BENCHMARK RESULTS

**Sth-Else Experiments.** Table 1 shows two experiments against benchmarks for all-way-k-shot learning using the ViT-B/16 backbone for the Sth-Else dataset. In the first experiment (top) we compare $TCMT_H$ to leading benchmarks on Sth-Else that use cross-entropy loss, ORViT and SViT (Herzig et al., 2022; Ben Avraham et al., 2022). In the second experiment (bottom) we compare $TCMT_C$ to the state-of-the-art methods with vision-language learning, ViFi-CLIP and VL-Prompting (Rasheed et al., 2023). Observe that $TCMT_C$ and $TCMT_H$ had the highest accuracy in both experiments. TCMT outperforms all four leading benchmarks by amounts ranging from 1.4 to 3.6 percentage points. For this dataset we see the greatest improvement for $k = 5$, improving from 34.4 to 37.6 and from 44.9 to 48.5 over the leading benchmarks, with smaller but significant improvements for $k = 10$. Also, Fig. 4 further shows that TCMT requires a smaller number of parameters to be updated during transfer.

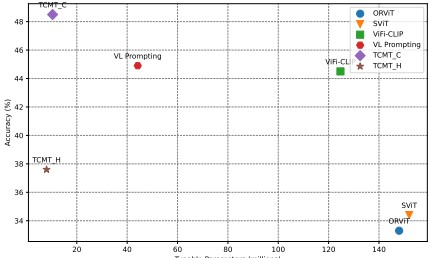

|  | Backbone | Sth-Else | |
|---|---|---|---|
|  |  | 5-shot | 10-shot |
| ORViT | ViT-B/16 | 33.3 | 40.2 |
| SViT | ViT-B/16 | 34.4 | 42.6 |
| **TCMT**$_H$ | ViT-B/16 | **37.6** | **44.0** |
| ViFi-CLIP | ViT-B/16 | 44.5 | 54.0 |
| VL Prompting | ViT-B/16 | 44.9 | 58.2 |
| **TCMT**$_C$ | ViT-B/16 | **48.5** | **59.9** |

Table 1: Comparing TCMT to benchmarks for all-way-k-shot learning on the Sth-Else dataset.

Figure 4: Comparing model performance for all-way-k-shot learning with $k = 5$ on the Sth-Else dataset versus number of parameters updated during adaptation.

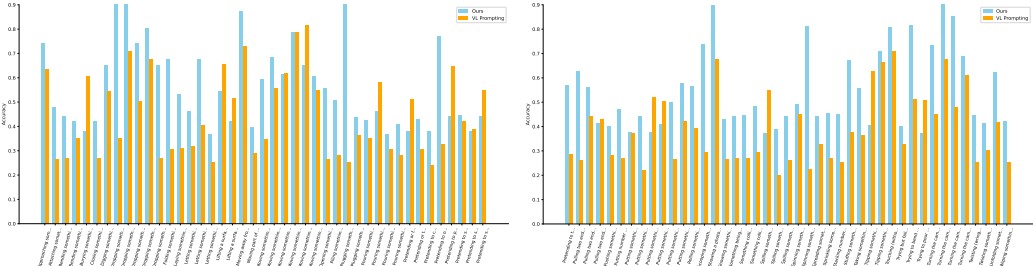

Figure 5: Comparing performance of $TCMT_C$ against VL-Prompting across all action classes on the Sth-Else dataset.

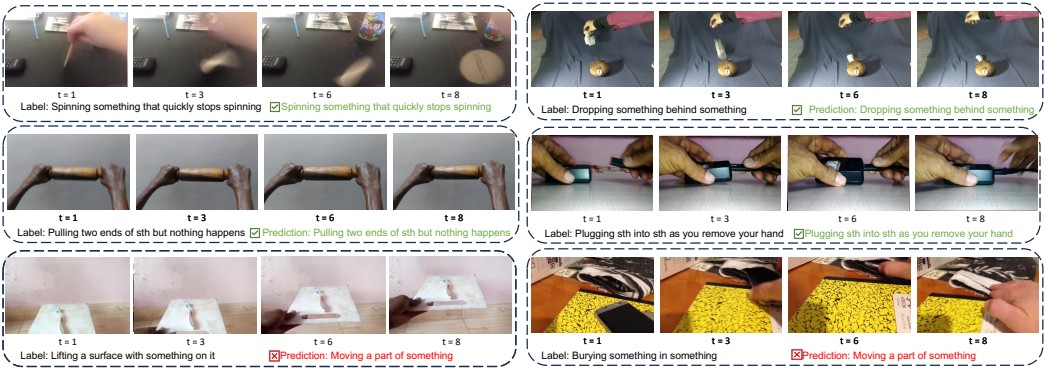

Figure 6: TCMT predictions on six example videos from the Sth-Else dataset.

Fig. 5 shows that TCMT outperforms VL-Prompting on the majority of action classes. Notably, TCMT exceeds the VL prompt in 72 out of 86 action classes, such as "plugging something into something" and "spinning so it continues spinning." We also note significant improvements in other categories, including "approaching something with something" and "pulling two ends of something but nothing happens". However, TCMT does have some limitations. Specifically, VL-Prompting surpasses our method in label classes like "burying something in something" and "lifting a surface with something on it".

Fig. 6 shows several examples in which $TCMT_C$ correctly predicts the four actions but misclassifies two videos of "lifting a surface with something on it", and "Burying something in something" as "Moving a part of something".

**All-way-k-shot Experiments.** Table 2 and Table 3 show the experimental results using the ViT-B/16 backbone comparing $TCMT_C$ to leading benchmarks for all-way-k-shot learning for $k \in \{2, 4, 8, 16\}$. For all-way-k-shot learning, our benchmarks are: ActionCLIP (Wang et al., 2021), XCLIP and XFLORENCE (Ni et al., 2022), VideoPrompt (Ju et al., 2022), VL Prompting and ViFi-CLIP (Rasheed et al., 2023), and VicTR (Kahatapitiya et al., 2023).

Table 2: Comparing $TCMT_C$ to benchmarks for all-way-k-shot on the HMDB-51 and UCF-101 datasets.

| | | HMDB-51 | | | | UCF-101 | | | |
|---|---|---|---|---|---|---|---|---|---|
| | Backbone | 2-shot | 4-shot | 8-shot | 16-shot | 2-shot | 4-shot | 8-shot | 16-shot |
| ActionCLIP | ViT-B/16 | 47.5 | 57.9 | 57.3 | 59.1 | 70.6 | 71.5 | 73.0 | 91.4 |
| XCLIP | ViT-B/16 | 53.0 | 57.3 | 62.8 | 64.0 | 70.6 | 71.5 | 73.0 | 91.4 |
| VideoPrompt | ViT-B/16 | 39.7 | 50.7 | 56.0 | 62.4 | 71.4 | 79.9 | 85.7 | 89.9 |
| ViFi-CLIP | ViT-B/16 | 57.2 | 62.7 | 64.5 | 66.8 | 80.7 | 85.1 | 90.0 | 92.7 |
| VL Prompting | ViT-B/16 | _63.0_ | _65.1_ | _69.6_ | _72.0_ | **91.0** | _93.7_ | _95.0_ | _96.4_ |
| VicTR | ViT-B/16 | 60.0 | 63.2 | 66.6 | 70.7 | 87.7 | 92.3 | 93.6 | 95.8 |
| **$TCMT_C$** | ViT-B/16 | **65.8** | **70.2** | **72.5** | **75.7** | _90.6_ | **94.7** | **96.2** | **98.5** |

Table 3: Comparing $TCMT_C$ to benchmarks for all-way-k-shot on the SSv2 dataset.

| | | SSv2 | | | |
|---|---|---|---|---|---|
| | Backbone | 2-shot | 4-shot | 8-shot | 16-shot |
| ActionCLIP | ViT-B/16 | 4.1 | 5.8 | 8.4 | 11.1 |
| XCLIP | ViT-B/16 | 3.9 | 4.5 | 6.8 | 10.0 |
| XFLORENCE | ViT-B/16 | 4.2 | 6.1 | 7.9 | 10.4 |
| VideoPrompt | ViT-B/16 | 4.4 | 5.1 | 6.1 | 9.7 |
| ViFi-CLIP | ViT-B/16 | 6.2 | 7.4 | 8.5 | 12.4 |
| VL Prompting | ViT-B/16 | _6.7_ | _7.9_ | _10.2_ | _13.5_ |
| **$TCMT_C$** | ViT-B/16 | **7.5** | **9.6** | **11.8** | **15.5** |

We follow Wang et al. (2021); Ni et al. (2022); Ju et al. (2022); Rasheed et al. (2023) in using Kinetics-400 (K-400) from (Carreira & Zisserman, 2017) as the base dataset $\mathcal{D}$ and repeat the experiment using novel data from the SSV2, HMDB-51, and UCF-101 datasets. Observe that $TCMT_C$ had the highest accuracy in 10 out of 11 of these experiments, and had the second highest accuracy in the remaining experiment trailing by only 0.4.

**5-way-k-shot Experiments.** Table 6 shows the results of experiments comparing $TCMT_H$ to leading benchmarks for 5-way-k-shot learning. All models are trained on K-400 as the base data using the ResNet-50 backbone. In each trial, we select 5 action classes at random and update our model using only $k$ samples for each of those classes to form $\mathcal{S}$, and the remaining novel data compose $\mathcal{Q}$. To ensure the statistical significance, we conduct 10000 trials with random samplings for selecting the action classes in each trial by following (Wang et al., 2023e), and report the mean accuracy as the final result. Our leading benchmarks are: OTAM (Cao et al., 2020), TRX (Perrett et al., 2021), STRM (Thatipelli et al., 2022), DYDIS (Islam et al., 2021), STARTUP (Phoo & Hariharan, 2021), and SEEN (Wang et al., 2023e). Observe that $TCMT_H$ has the highest accuracy in 7 out of 8 experiments, by a margin ranging from 1.2 to 5.3, and has the second highest accuracy on the remaining experiment, trailing by only 0.1.

Table 4: Comparing $\text{TCMT}_H$ to benchmarks for 5-way-k-shot learning.

|  | Backbone | 1-shot | | | | 5-shot | | | |
|---|---|---|---|---|---|---|---|---|---|
|  |  | UCF-101 | HMDB-51 | SSv2 | SSv2-small | UCF-101 | HMDB-51 | SSv2 | SSv2-small |
| OTAM | ResNet-50 | 50.2 | 34.4 | 24.0 | 22.4 | 61.7 | 41.5 | 27.1 | 25.8 |
| TRX | ResNet-50 | 47.1 | 32.0 | 23.2 | 22.9 | 66.7 | 43.9 | 27.9 | 26.0 |
| STRM | ResNet-50 | 49.2 | 33.0 | 23.6 | 22.8 | 67.0 | 45.2 | 28.7 | 26.4 |
| DYDIS | ResNet-50 | 63.4 | 35.2 | 25.3 | 24.8 | 77.5 | 50.8 | 29.3 | 27.2 |
| STARTUP | ResNet-50 | 65.4 | 35.5 | 25.1 | 25.0 | 79.5 | 50.4 | 31.3 | 28.7 |
| SEEN | ResNet-50 | 64.8 | 35.7 | 26.1 | 25.3 | **79.8** | 51.1 | 34.4 | 29.3 |
| **TCMT**$_H$ | ResNet-50 | **66.0** | **37.6** | **31.4** | **28.5** | 79.7 | **54.4** | **37.0** | **32.8** |

## 3.3 Ablation Study

We run an ablation study to select the hyperparameters of our model. We compare $\text{TCMT}_C$ with the different numbers of latent causal variables ($N \in \{4, 8, 12, 16\}$) using the ViT-B/16 backbone for all-way-5-shot and all-way-10-shot learning on the Sth-Else data. We can see that performance is increasing in $N$, verifying that the latent causal variables facilitate few-shot action recognition. Based on the results in Table 5 we choose $N = 12$ because beyond this point the performance gain becomes marginal and the computational cost spikes.

Once $N = 12$ was selected, we proceeded by comparing our TCMT model to four simpler models; one non-causal, one non-temporal, one without auxiliary context variables $\theta$, and one with fixed ConvLSTM on both $\mathcal{D}$ and $\mathcal{S}$, all with $N = 12$. In the non-causal model the time-delayed causal transitions between latent variables (red arrows in Fig. 3) are removed so that $\mathbf{x}$ and $\mathbf{y}$ are no longer independent conditioned on $\mathbf{z}$, and the temporal posterior is regularized by KL-divergence with the standard normal distribution without using our prior network. In the non-temporal model the latent variables do not change over time ($\mathbf{z}_t = \mathbf{z}_{t+1} \forall t$), but it is still a causal model so $\mathbf{x}$ and $\mathbf{y}$ are independent conditioned on $\mathbf{z}$. As with the full TCMT model, in the non-temporal model the posterior is regularized by KL-divergence with the prior output by the prior network. The last second line in Table 5 gives a comparison to a model without any auxiliary context variables. We also assess the efficacy of fixing the transition and mixing functions versus fine-tuning them (TCMT-FT). The observation that TCMT significantly outperforms these three baselines motivates our temporal causal mechanism with updating auxiliary context variables on $\mathcal{Q}$.

Notice that the accuracies of 46.8 achieved by $\text{TCMT}_C$ without $\theta$ and 45.1 by the non-temporal model for all-way-5-shot learning in our ablation study are already better than the 44.5 and 44.9 achieved by ViFi-CLIP and VL-prompting, respectively (Rasheed et al., 2023) (Table 1). This demonstrates the transportability of the causal mechanism. Moreover, our temporal causal model performs even better because it captures the temporal relations of the latent causal variables, and $\theta$ helps capture the distribution shift across data sets.

Table 5: Ablation study for selecting hyperparameters on the Sth-Else dataset.

|  | Backbone | 5-shot | 10-shot |
|---|---|---|---|
| $N = 4$ | ViT-B/16 | 44.2 | 49.0 |
| $N = 8$ | ViT-B/16 | 46.0 | 55.7 |
| $N = 12$ | ViT-B/16 | 48.5 | 59.9 |
| $N = 16$ | ViT-B/16 | **48.9** | **60.1** |
| Non-causal | ViT-B/16 | 41.0 | 44.3 |
| TCMT-FT | ViT-B/16 | 44.8 | 55.3 |
| Non-temporal | ViT-B/16 | 45.1 | 54.4 |
| Without $\theta$ | ViT-B/16 | 46.8 | 57.6 |

## 4 Related Work

**Few-shot Action Recognition.** Few-shot learning has become an important research area in computer vision because we often want to identify classes with only a small number of labeled samples available Chen et al. (2019); Tseng et al. (2020); Luo et al. (2023). Metric-based techniques are popular for many few-shot classification problems. These metric-based methods use an encoder to map visual data into an embedding space and learn a similarity (or distance) function, e.g. cosine distance Vinyals et al. (2016). For example, ProtoNet (Snell et al., 2017) learns a feature extractor to transform all samples into a common feature space and treats the mean features of support images as prototypes for matching. The key assumption here is generally that the embedding space is common to the base and novel datasets. Gradient optimization techniques such as MAML are also common Finn et al. (2017).

Compared to images, video data adds a temporal dimension rich with spatio-temporal information and variation Poppe (2010). Some of our benchmarks use metric-based approaches for few-shot action recognition on video data. OTAM offers a differentiable dynamic time warping algorithm Cao et al. (2020), TRX uses an attention mechanism Perrett et al. (2021), and STRM introduces a joint patio-temporal modeling technique Thatipelli et al. (2022). A few studies try to use meta-learning to deal with the temporal nature of action recognition (Finn et al., 2017). For instance, STRM (Thatipelli et al., 2022) introduces a spatio-temporal enrichment module to look at visual and temporal context at the patch and frame level. HyRSM (Wang et al., 2022b) uses a hybrid relation model to learn relations within and across videos in a given few-shot episode. However, recent work has shown that these methods do not generalize well when the base and novel data have significant distribution disparities (Wang et al., 2023e; Samarasinghe et al., 2023).

The recent success of cross-modal vision-language learning has inspired works like ActionCLIP (Wang et al., 2021), XCLIP and XFLORENCE (Ni et al., 2022), VideoPrompt (Ju et al., 2022), VL Prompting and ViFi-CLIP (Rasheed et al., 2023). All of these methods apply transfer learning by adopting the pre-trained CLIP and adapting it for few-shot action recognition tasks. These methods utilize the rich generalized representations of CLIP and fuse them with additional components for temporal modeling. Nevertheless, if the CLIP model is pre-trained on a base dataset that is too dissimilar from the novel data, the issues from distribution disparity persist. Unlike previous work, TCMT approaches few-shot action recognition by leveraging causal representation learning to develop generative models of temporal causal processes, which our experiments show provide an advantage for handling distribution disparities.

**Causal Representation Learning.** TCMT is based on causal representation learning (Khemakhem et al., 2020). Several recent works have extended the theory of causal representation learning to different fields, such as domain adaption (Teshima et al., 2020; Kong et al., 2022), domain generalization (Xie et al., 2023) and medical image analysis (Wang et al., 2023a). Additionally, some works try to model temporal causal representations for time series data. For instance, the authors of (Huang et al., 2022; Feng et al., 2022) introduce causal representation to Markov decision processes (MDP). (Yao et al., 2022b;a) provides the theoretical proof of the identifiability of causal representations from time-series data. Based on these identifiability results, TCMT applies causal representation learning to time series data in the context of few-shot action recognition.

We also mention a line of work based on causal inference (Pearl et al., 2009; Rubin, 2019). This line of work also tries to learn invariant features, but usually relies on the assumption that the causal graph structure is known beforehand, and leverages the potential outcome framework, grounded on this assumed causal graph, for causal effect estimation (Wang et al., 2020; Yue et al., 2020; Niu et al., 2021; Chen et al., 2023). Also, the authors of (Wang et al., 2022a;c; Liu et al., 2021; Lin et al., 2022; Yi et al., 2022) implicitly assume the invariant features to solve the invariant risk minimization problem. By contrast, our method endeavors to uncover the true causal structure, anchoring our TCMT framework in the domain of causal representation learning.

# 5 CONCLUSION

We propose Temporal Causal Mechanism Transfer (TCMT) for few-shot action recognition, which relies on variational inference to learn a temporal causal mechanism from base data that can be efficiently and effectively adapted to novel data by few-shot learning. We demonstrate that our approach is able to achieve accuracy comparable to or better than leading benchmarks on a variety of standard video sequence with fewer parameter updates during the adaptation to novel data.

We note that TCMT has two major limitations. First, the transitions between latent causal variables are assumed to be strictly time-delayed with no instantaneous causal relations. While this corresponds to our intuitive notions of causality, this assumption will be violated in time series data if the observed time frequency is low. Second, it is difficult to infer the auxiliary context variables $\theta$ from the real-world video sequence, and we provide only a coarse way to model it. A better model of $\theta$ could improve the performance of TCMT. Extending TCMT to address instantaneous causal relations and providing a better model of the auxiliary context variables are clear directions for future work. Lastly, we mention that exploring the merits of temporal causal mechanisms for LLM agents is an interesting direction (Yao et al., 2023; Wang et al., 2023b).

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

## A APPENDIX

### A.1 TRANSITION PRIOR LIKELIHOOD DERIVATION

Consider a paradigmatic instance of latent causal processes. In this case, we are concerned with two time-delayed latent variables, namely, $\mathbf{z}_t = [\mathbf{z}_{1,t}, \mathbf{z}_{2,t}]$. Crucially, there is no inclusion of $\boldsymbol{\theta}$. The maximum time lag is defined as 1. This implies that each latent variable, $\mathbf{z}_{n,t}$, is formulated as $\mathbf{z}_{n,t} = f_n(\mathbf{z}_{t-1}, \epsilon_{n,t})$, where the noise terms, $\epsilon_{n,t}$, are mutually independent. To represent this latent process more succinctly, we introduce a transformation map, denoted as $f$. It's worth noting that in this context, we employ an overloaded notation; specifically, the symbol $f$ serves dual purposes, representing both transition functions and the transformation map.

$$\begin{bmatrix} \mathbf{z}_{1,t-1} \\ \mathbf{z}_{2,t-1} \\ \mathbf{z}_{1,t} \\ \mathbf{z}_{2,t} \end{bmatrix} = \mathbf{f}\left( \begin{bmatrix} \mathbf{z}_{1,t-1} \\ \mathbf{z}_{2,t-1} \\ \epsilon_{1,t} \\ \epsilon_{2,t} \end{bmatrix} \right). \tag{15}$$

By leveraging the change of variables formula on the map $\mathbf{f}$, we can evaluate the joint distribution of the latent variables $p(\mathbf{z}_{1,t-1}, \mathbf{z}_{2,t-1}, \mathbf{z}_{1,t}, \mathbf{z}_{2,t})$ as:

$$p(\mathbf{z}_{1,t-1}, \mathbf{z}_{2,t-1}, \mathbf{z}_{1,t}, \mathbf{z}_{2,t}) = p(\mathbf{z}_{1,t-1}, \mathbf{z}_{2,t-1}, \epsilon_{1,t}, \epsilon_{2,t})/\left| \det \mathbf{J}_{\mathbf{f}} \right|, \tag{16}$$

where $\mathbf{J}_{\mathbf{f}}$ is the Jacobian matrix of the map $\mathbf{f}$, which is naturally a low-triangular matrix:

$$\mathbf{J}_{\mathbf{f}} = \begin{bmatrix} 1 & 0 & 0 & 0 \\ 0 & 1 & 0 & 0 \\ \frac{\partial \mathbf{z}_{1,t}}{\partial \mathbf{z}_{1,t-1}} & \frac{\partial \mathbf{z}_{1,t}}{\partial \mathbf{z}_{2,t-1}} & \frac{\partial \mathbf{z}_{1,t}}{\partial \epsilon_{1,t}} & 0 \\ \frac{\partial \mathbf{z}_{2,t}}{\partial \mathbf{z}_{1,t-1}} & \frac{\partial \mathbf{z}_{2,t}}{\partial \mathbf{z}_{2,t-1}} & 0 & \frac{\partial \mathbf{z}_{2,t}}{\partial \epsilon_{2,t}} \end{bmatrix}.$$

Given that this Jacobian is triangular, we can efficiently compute its determinant as $\prod_n \frac{\partial \mathbf{z}_{n,t}}{\partial \epsilon_{n,t}}$. Furthermore, because the noise terms are mutually independent, and hence $\epsilon_{n,t} \perp \epsilon_{l,t}$ for $m \neq n$ and $\epsilon_t \perp \mathbf{z}_{t-1}$, we can write Eq. 16 as:

$$\begin{aligned} p(\mathbf{z}_{1,t-1}, \mathbf{z}_{2,t-1}, \mathbf{z}_{1,t}, \mathbf{z}_{2,t}) &= p(\mathbf{z}_{1,t-1}, \mathbf{z}_{2,t-1}) \times p(\epsilon_{1,t}, \epsilon_{2,t})/\left| \det \mathbf{J}_{\mathbf{f}} \right| \quad (\text{because } \epsilon_t \perp \mathbf{z}_{t-1}) \\ &= p(\mathbf{z}_{1,t-1}, \mathbf{z}_{2,t-1}) \times \prod_n p(\epsilon_{n,t})/\left| \det \mathbf{J}_{\mathbf{f}} \right| \quad (\text{because } \epsilon_{1,t} \perp \epsilon_{2,t}) \end{aligned} \tag{17}$$

By eliminating the marginals of the lagged latent variable $p(\mathbf{z}_{1,t-1}, \mathbf{z}_{2,t-1})$ on both sides, we derive the transition prior likelihood as:

$$p(\mathbf{z}_{1,t}, \mathbf{z}_{2,t}|\mathbf{z}_{1,t-1}, \mathbf{z}_{2,t-1}) = \prod_n p(\epsilon_{n,t})/\left| \det \mathbf{J}_{\mathbf{f}} \right| = \prod_n p(\epsilon_{n,t}) \times \left| \det \mathbf{J}_{\mathbf{f}}^{-1} \right|. \tag{18}$$

Let $\{f_n^{-1}\}_{n=1,2,3\ldots}$ be a set of learned inverse dynamics transition functions that take the estimated latent causal variables in the fixed dynamics subspace and lagged latent variables, and output the noise terms, i.e., $\hat{\epsilon}_{n,t} = f_n^{-1}(\hat{\mathbf{z}}_{n,t}, \text{Pa}(\hat{\mathbf{z}}_{n,t}))$.

The differences of our model from Eq. 18 are that the learned inversedynamics transition functions take $\theta$ as input arguments to out the noise terms, i.e, $\hat{\epsilon}_{n,t} = f_n^{-1}(\hat{\mathbf{z}}_{n,t}, \text{Pa}(\hat{\mathbf{z}}_{n,t}), \boldsymbol{\theta}_t) = f_n^{-1}(\hat{\mathbf{z}}_{n,t}, \text{Pa}(\hat{\mathbf{z}}_{n,t}), \boldsymbol{\theta}_t)$.

$$\log p(\hat{\mathbf{z}}_t|\hat{\mathbf{z}}_{t-1}, \boldsymbol{\theta}_t) = \sum_{n=1}^{N} \log p(\hat{\epsilon}_{n,t}|\boldsymbol{\theta}_t) + \sum_{n=1}^{N} \log \left| \frac{\partial f_n^{-1}}{\partial \hat{\mathbf{z}}_{n,t}} \right| \tag{19}$$

## A.2 ADDITIONAL EXPERIMENTS

**Dataset details.** In this paper, we detail experiments conducted on five datasets:: 1. Something-Something v2 (SSv2) is a dataset containing 174 action categories of common human-object inter-actions; 2. Something-Else (Sth-Else) exploits the compositional structure of SSv2, where a com-bination of a verb and a noun defines an action; 3. HMDB-51 contains 7k videos of 51 categories; 4. UCF-101 covers 13k videos spanning 101 categories; 5. Kinetics covers around 230k 10-second video clips sourced from YouTube.

**Additonal experiments.** In this section, we focus on comparisons with state-of-the-art metric-based methods, including MoLo (Wang et al., 2023d), HySRM (Wang et al., 2022b), HCL (Zheng et al., 2022), OTAM (Cao et al., 2020), TRX (Perrett et al., 2021) and STRM (Thatipelli et al., 2022). For a fair evaluation, we perform experiments across the SSv2, UCF-101, HMDB-51, and Kinetics datasets. In the SSv2-Full and SSv2-Small datasets, we randomly selected 64 classes for $\mathcal{D}$ and 24 for $\mathcal{S}$ and $\mathcal{Q}$. The main difference between SSv2-Full and SSv2-Small is the dataset size, with SSv2-Full containing all samples per category and SSv2-Small including only 100 samples per category. For HMDB-51, we chose 31 action categories for $\mathcal{D}$ and 10 for $\mathcal{S}$ and $\mathcal{Q}$, while for UCF-101, the selection was 70 and 21 categories, respectively. For Kinect, we used 64 action categories for $\mathcal{D}$ and 24 for $\mathcal{S}$ and $\mathcal{Q}$. To maintain statistical significance, we executed 200 trials, each involving random samplings across categories. After training on $\mathcal{D}$, we used $k$ video sequences from each action category to form $\mathcal{S}$ for model updates. The inference phase utilized the remaining data from $\mathcal{Q}$.

Table 6 and Table 7 show additional experiments for 5-way-k-shot learning where the base and novel data are taken from the same original dataset. Between the two tables we osberve that TCMT$_H$ achieves the highest Top-1 accuracy in 11 out of 12 experiments, coming in second by only $0.4$ with $k = 5$ on the UCF-101 dataset.

Table 6: Comparing TCMT$_H$ to benchmarks for 5-way-k-shot learning on the SSv2 and SSv2-small

|  |  | SSv2 | | | SSv2-small | | |
| --- | --- | --- | --- | --- | --- | --- | --- |
|  | Backbone | 1-shot | 3-shot | 5-shot | 1-shot | 3-shot | 5-shot |
| OTAM | ResNet-50 | 42.8 | 51.5 | 52.3 | 36.4 | 45.9 | 48.0 |
| TRX | ResNet-50 | 42.0 | 57.6 | 62.6 | 36.0 | 51.9 | 56.7 |
| STRM | ResNet-50 | 42.0 | 59.1 | 68.1 | 37.1 | 49.2 | 55.3 |
| HyRSM | ResNet-50 | 54.3 | 65.1 | 69.0 | 40.6 | 52.3 | 56.1 |
| HCL | ResNet-50 | 47.3 | 59.0 | 64.9 | 38.7 | 49.1 | 55.4 |
| MoLo | ResNet-50 | 56.6 | 67.0 | 70.6 | 42.7 | 52.9 | 56.4 |
| **TCMT$_H$** | ResNet-50 | **60.0** | **68.3** | **71.9** | **45.8** | **53.6** | **58.0** |

Table 7: Comparing TCMT$_H$ to benchmarks for 5-way-k-shot learning on the UCF-101, HMDB-51, and Kinectics datasets.

|  |  | UCF-101 | | HMDB-51 | | Kinects | |
| --- | --- | --- | --- | --- | --- | --- | --- |
|  | Backbone | 1-shot | 5-shot | 1-shot | 5-shot | 1-shot | 5-shot |
| OTAM | ResNet-50 | 79.9 | 88.9 | 54.5 | 68.0 | 79.9 | 88.9 |
| TRX | ResNet-50 | 78.2 | 96.1 | 53.1 | 75.6 | 78.2 | 96.2 |
| STRM | ResNet-50 | 80.5 | **96.9** | 52.3 | 77.3 | 80.5 | 96.9 |
| HyRSM | ResNet-50 | 83.9 | 94.7 | 60.3 | 76.0 | 83.9 | 94.7 |
| HCL | ResNet-50 | 82.8 | 93.3 | 59.1 | 76.3 | 73.7 | 85.8 |
| MoLo | ResNet-50 | 86.0 | 95.5 | 60.8 | 77.4 | 86.0 | 95.5 |
| **TCMT$_H$** | ResNet-50 | **87.3** | 96.5 | **61.9** | **80.5** | **86.1** | **98.0** |

## A.3 COMPARISONS TO AUGMENTED MODELS

We further assess if adding $\theta$ to other methods would further improve the results in Table 8. The results indicate marginal improvements over the original methods. Again, our TCMT obtains the highest accuracy among these methods.

Table 8: Additional comparisons by augmenting existing methods on the Sth-Else dataset. $+\theta$ means the method updates the Context Network when adapting instead of fine-tuning. Since VL Prompting uses VPT (Jia et al., 2022) within the ViFi-CLIP framework, we only test ViFi-CLIP$+\theta$.

|  |  | Sth-Else | |
| --- | --- | --- | --- |
|  | Backbone | 5-shot | 10-shot |
| ORViT | ViT-B/16 | 33.3 | 40.2 |
| ORViT $+ \theta$ | ViT-B/16 | 33.9 | 41.8 |
| SViT | ViT-B/16 | 34.4 | 42.6 |
| SViT $+ \theta$ | ViT-B/16 | 35.2 | **44.0** |
| **TCMT**$_H$ | ViT-B/16 | **37.6** | **44.0** |
| ViFi-CLIP | ViT-B/16 | 44.5 | 54.0 |
| VL Prompting | ViT-B/16 | 44.9 | 58.2 |
| ViFi-CLIP $+ \theta$ | ViT-B/16 | 45.2 | 58.0 |
| **TCMT**$_C$ | ViT-B/16 | **48.5** | **59.9** |

## A.4 ADDITIONAL ABLATION EXPERIMENTS

We validate the performance of TCMT$_C$ compared to TCMT-FT on the SSv2, HMDB-51, and UCF-101 datasets under all-way-k-shot settings. Tables 9 and 10 show that by maintaining fixed transition and mixing functions, TCMT$_C$ attains superior scores. This outcome underscores the generalizability of our assumption regarding the temporal causal mechanism transfer.

Table 9: Comparing TCMT$_C$ to TCMT-FT for all-way-k-shot on the SSv2 dataset.

|  |  | SSv2 | | | |
| --- | --- | --- | --- | --- | --- |
|  | Backbone | 2-shot | 4-shot | 8-shot | 16-shot |
| TCMT-FT | ViT-B/16 | 6.1 | 7.9 | 10.4 | 14.1 |
| TCMT$_C$ | ViT-B/16 | 7.5 | 9.6 | 11.8 | 15.5 |

Table 10: Comparing TCMT$_C$ to TCMT-FT for all-way-k-shot on the HMDB-51 and UCF-101 datasets.

|  |  | HMDB-51 | | | | UCF-101 | | | |
| --- | --- | --- | --- | --- | --- | --- | --- | --- | --- |
|  | Backbone | 2-shot | 4-shot | 8-shot | 16-shot | 2-shot | 4-shot | 8-shot | 16-shot |
| TCMT-FT | ViT-B/16 | 61.5 | 64.8 | 70.0 | 72.9 | 87.7 | 93.7 | 95.1 | 96.4 |
| TCMT$_C$ | ViT-B/16 | 65.8 | 70.2 | 72.5 | 75.7 | 90.6 | 94.7 | 96.2 | 98.5 |

For each data sample we must select a number of frames to use from a video sequence. We extend our ablation study for selecting the number of frames. In Table 11 we find that 8 frames per video is sufficient, with marginal improvement for larger numbers of frames.

In our main experiments, the parents of each latent causal variable are only in the previous time step. Table 12 reports that we allow the parents to be further back in time, allowing causal relationships between non-consecutive time steps, and show that there is a negligible improvement in accuracy.

In Table 13, we compare our method with an alternative method that first trains the auto-encoder and context network and then the classifier separately. We observe no significant difference in performance on the Sth-Else dataset.

## A.5 MOTIVATING EXAMPLES FOR OTHER DATASETS

Figure 7, Figure 8 and Figure 9 showcase the similar motivating examples of a few-shot action recognition task with TCMT in using SSv2, HMDB-51 and UCF-101 as novel data, while the K-400 serves as base data. Please see the caption for the detailed explanations.

Table 11: Ablation study for selecting the different length of input on the Sth-Else dataset.

| input frames | Backbone | 5-shot | 10-shot |
|---|---|---|---|
| 4 | ViT-B/16 | 43.3 | 51.0 |
| 8 | ViT-B/16 | 48.5 | 59.9 |
| 16 | ViT-B/16 | 50.8 | 61.2 |

Table 12: Ablation study allowing parents of $z$ to be in $z_{t-1}$ through $z_{t-\tau}$ for $\tau \in 1, 2, 3$.

| | Backbone | 5-shot | 10-shot |
|---|---|---|---|
| $\tau = 1$ | ViT-B/16 | 48.5 | 59.9 |
| $\tau = 2$ | ViT-B/16 | 48.5 | 60.0 |
| $\tau = 3$ | ViT-B/16 | 48.8 | 60.2 |

## A.6 NETWORK ARCHITEXTURES

Tab. 14 illustrates the details of our implementation on $\text{TCMT}_C$.

## A.7 CODE

Our code will be released upon the acceptance.

Table 13: Ablation study for separate training vs. joint training.

|  | 5-shot | 10-shot |
|---|---|---|
| Joint training | 48.5 | 59.9 |
| Separate training | 48.5 | 60.0 |

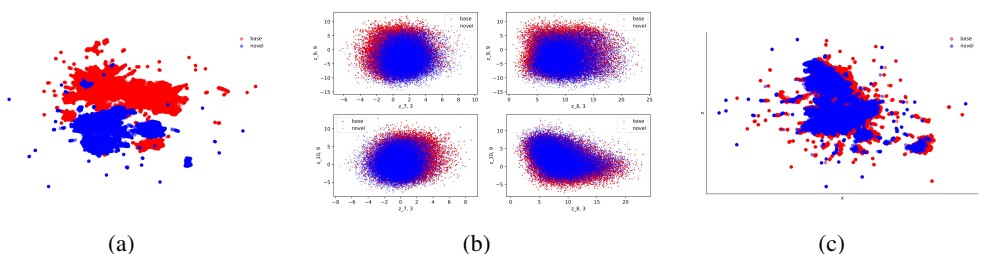

(a)                    (b)                    (c)

Figure 7: Motivating example of Temporal Causal Mechanism Transfer (TCMT) for few-shot action recognition in using Kinetics-400 as the base dataset $\mathcal{D}$ and SSV2 as novel data (*The original PDF files for this Figure are available in zip of supplementary, providing additional and clearer illustration*): (a) shows the UMap visualization of the action feature embeddings obtained by a fixed ViT-B/16 backbone (Radford et al., 2021) on the base data (red) and novel data (blue). The obvious distribution disparities demonstrate the difficulty of few-shot learning for action recognition; (b) shows an example of pairplot of the values of latent causal variables from the transition functions of two trained models, one trained only on the base data (red) and the other only on the novel data (blue). (c) shows a pairplot of UMap projections for action feature embeddings $x$ and latent variables $z$ from two models: one trained on base data (red) and the other on novel data (blue). The near-perfect alignment in (b) and (c) supports our hypothesis of invariant transition and mixing functions across base and novel data.

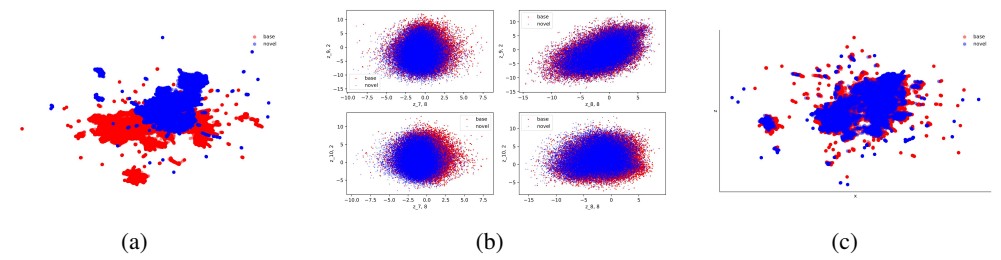

(a)                    (b)                    (c)

Figure 8: Motivating example of Temporal Causal Mechanism Transfer (TCMT) for few-shot action recognition in using K-400 as base data and HMDB-51 as novel data (blue) (*The original PDF files for this Figure are available in zip of supplementary, providing additional and clearer illustration*): (a) shows the UMap visualization of the action feature embeddings obtained by a fixed ViT-B/16 backbone (Radford et al., 2021) on the base data (red) and novel data (blue). The distribution disparities demonstrate the difficulty of few-shot learning for action recognition; (b) shows an example of pairplot of the values of latent causal variables from two trained models, one trained only on the base data (red) and the other only on the novel data (blue). (c) shows a pairplot of UMap projections for action feature embeddings $x$ and latent variables $z$ from two models: one trained on base data (red) and the other on novel data (blue). The near-perfect alignment in (b) and (c) supports our hypothesis of invariant transition and mixing functions across base and novel data.

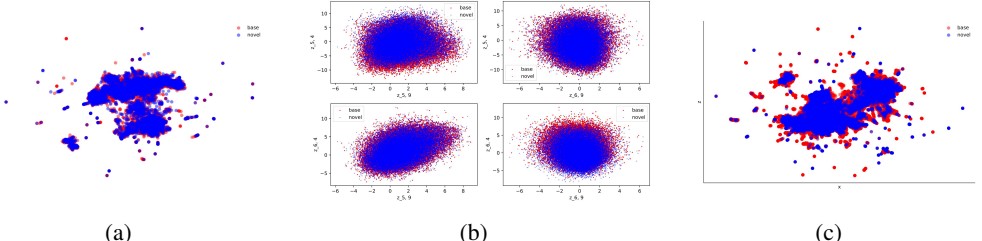

|  (a) | (b) | (c) |
| --- | --- | --- |

Figure 9: Motivating example of Temporal Causal Mechanism Transfer (TCMT) for few-shot action recognition in using Kinetics-400 as the base dataset $\mathcal{D}$ and the UCF-101 as the novel data (*The original PDF files for this Figure are available in zip of supplementary, providing additional and clearer illustration*): (a) shows the UMap visualization of the action feature embeddings obtained by a fixed ViT-B/16 backbone (Radford et al., 2021) on the base data (red) and novel data (blue). The distribution alignment is consistent with the high accuracy obtained in Table 2. (b) shows an example of pairplot of the values of latent causal variables of the transition functions from two trained models, one trained only on the base data (red) and the other only on the novel data (blue). (c) shows a pairplot of UMap projections for action feature embeddings $x$ and latent variables $z$ from two models: one trained on base data (red) and the other on novel data (blue). The near-perfect alignment in (b) and (c) supports our hypothesis of invariant transition and mixing functions across base and novel data.

Table 14: The details of our network architectures for TCMT$_C$, where BS means batch size.

| Configuration | Description | Output dimensions |
| --- | --- | --- |
| **Encoder** | | |
| Input: concat($\mathbf{x}_{1:T}, \boldsymbol{\theta}_{1:T}$) | | BS $\times T \times (1024 + 128)$ |
| Dense | 256 neurons, LeakyReLU | BS $\times T \times 256$ |
| Dense | 256 neurons, LeakyReLU | BS $\times T \times 256$ |
| Dense | Temporal embeddings | BS $\times T \times 2N$ |
| Bottleneck | Compute mean and variance of posterior | $\mu, \sigma$ |
| Reparameterization | Sequential sampling | $\hat{\mathbf{z}}_{1:T}$ |
| | | |
| **Decoder** | | |
| Input: $\hat{\mathbf{z}}_{1:T}$ | | BS $\times T \times N$ |
| Dense | 256 neurons, LeakyReLU | BS $\times T \times 256$ |
| Dense | 256 neurons, LeakyReLU | BS $\times T \times 256$ |
| Dense | input embeddings | BS $\times T \times 1024$ |
| | | |
| **Prior** | | |
| Input | $\hat{\mathbf{z}}_{1:T}$ | BS $\times T \times N$ |
| InverseTransition | $\epsilon_t$ | BS $\times T \times N$ |
| JacobianCompute | $\log \det |J|$ | BS |
| | | |
| **Classifier** | | |
| Input: Concat($\hat{\mathbf{z}}_{1:T}$) | | BS $\times T \times N$ |
| Dense | 256 neurons, LeakyReLU | BS $\times T \times 256$ |
| Dense | 256 neurons, LeakyReLU | BS $\times T \times 256$ |
| Dense | output embeddings | BS $\times T \times 1024$ |
| **Context network** | | |
| Input: $\mathbf{x}_{1:T}$ | | BS $\times T \times 1024$ |
| Convolutional LSTM | ConvLSTM | BS $\times T \times 128$ |

