# OpenReview forum: "Temporal Causal Mechanism Transfer for Few-shot Action Recognition"
_ICLR.cc/2024/Conference — Submitted to ICLR 2024_

### Official Review · Reviewer_K1pA · 2023-10-30

**Soundness:** 3 good
**Presentation:** 2 fair
**Contribution:** 3 good
**Rating:** 6
**Confidence:** 4

**Summary:**

This paper introduces temporal causal mechanism transfer (TCMT) for few-shot action recognition. It considers the action sequences from a generative model perspective. Specifically, it assumes that base and novel action videos share some common causal relationships. By learning these causal relationships, the model can work better with less training data (few-shot recognition). The overall causal learning framework is built as a variational autoencoder. After the training, only the encoder is kept to perform action recognition with the intermediate representations. The proposed TCMT is evaluated on benchmark datasets including UCF101, HMDB51, and SSv2.

**Strengths:**

1) The idea is easy to follow and modeling the causal relationship for few-shot action recognition is novel and reasonable
2) This paper proposed to model the causal relationship between hidden variables and action sequences. By learning the invariant part of the relationship, the parameters of few-shot action recognition model can be reduced since only the auxiliary variable is needed to be considered at each time step.
3) Comparison of non-causal and causal demonstrates the effectiveness of the proposed method.

**Weaknesses:**

1) In the introduction part, there are no red arrows in Figure 2. But the explanation in the second last paragraph is explaining it using red arrows, which makes the time-delayed causal relations confusing.
2) Based on the proposed causal modeling process, it seems only first-order dependency is modeled. However, the action sequences probably has high-order dependencies.
3) The comparison is incomplete, missing many recent work such as:
[1] Wang, Xiang, et al. "MoLo: Motion-augmented Long-short Contrastive Learning for Few-shot Action Recognition." Proceedings of the IEEE/CVF Conference on Computer Vision and Pattern Recognition. 2023.
[2] Zheng, Sipeng, Shizhe Chen, and Qin Jin. "Few-shot action recognition with hierarchical matching and contrastive learning." European Conference on Computer Vision. Cham: Springer Nature Switzerland, 2022.
[3] Wang, Xiang, et al. "Hybrid relation guided set matching for few-shot action recognition." Proceedings of the IEEE/CVF Conference on Computer Vision and Pattern Recognition. 2022.
4) There is no justification whether the causal relationship is learned correctly besides the performance improvement.
5) For the comparison number of parameters, all parameters besides the parameters in the adaption process should be counted since they are needed for inference.

**Questions:**

0) It is very slow to open and scroll the submitted document locally. Perhaps Figure 1 (b) has too many objects. I don’t know if this only happen on my site.
1) For equation (11), is the ratio of L_{ELBO} and L_{cls} 1:1?

2) Just for curiosity, does the hidden variable theta have interpretable meanings? If theta control certain aspects of the action generation process, it would be easier to justify the causal relationship.

3) To training the autoencoder, joint training may not be optimal. If the CVAE is firstly trained for causal modeling and then jointly trained for maximizing ELBO and classification, maybe the causal relationship can be better learned. In addition, the results from the first step can be used to verify if the causal relationship is correctly captured.

4) In Table 5, what is the “N” used for non-causal, non-temporal, and without theta?

**Details Of Ethics Concerns:**

No ethics concerns

---

> ### Author Response · Authors · 2023-11-16
> **Response to Reviewer K1pA P1**
>
> We sincerely appreciate the insightful comments and helpful suggestions. We have carefully considered your feedback and addressed each point in our response, aiming to clarify and enhance the understanding of our research.
>
> >**Weakness 1:** In the introduction part, there are no red arrows in Figure 2. But the explanation in the second last paragraph is explaining it using red arrows, which makes the time-delayed causal relations confusing.
>
> **Answer:** We have corrected the typo referring to red arrows and polish Figure 2. Thank you for pointing this out.
>
> >**Weakness 2:** Based on the proposed causal modeling process, it seems only first-order dependency is modeled. However, the action sequences probably has high-order dependencies.
>
> **Answer:** This is an excellent point. Higher-order dependencies are certainly possible. In response we have added new results on the Sth-Else dataset with models that account for causal relations between non-consecutive time steps in Table A. We observe that the improvements are marginal although we experience a considerable increase in computational costs with higher-order models; with four 2080Ti GPUs, it takes 20 hours to train with only first-order dependencies $(\tau=1)$ compared to 30 hours for second-order $(\tau=2)$ and 36 hours for third-order $(\tau=3)$. We note that the failure of higher-order models to yield a benefit may be due to the nature of the Sth-Else dataset, which primarily consists of 1-2 second video sequences. Due to the computation cost of large-scale pre-training, we are not sure whether it can be done before the 22nd Nov for all datasets. We will update the results at our earliest once we obtain the results. These results has been added to Table 10 in the appendix.
>
> **Table A:** Ablation study allowing parents of $z$ to be in $z_{t-1}$ through $z_{t-\tau}$ for $\tau \in {1,2,3}$.
> |  |  5-shot | 10-shot |
> |--------|:--------:|:--------:|
> | $\tau=1$ | 48.5 | 59.9 |
> | $\tau=2$ | 48.5 | 60.0 |
> | $\tau=3$ | 48.8 | 60.2 |

---

> ### Author Response · Authors · 2023-11-16
> **Response to Reviewer K1pA P2**
>
> >**Weakness 3:** The comparison is incomplete, missing many recent work such as: [1] Wang, Xiang, et al. "MoLo: Motion-augmented Long-short Contrastive Learning for Few-shot Action Recognition." Proceedings of the IEEE/CVF Conference on Computer Vision and Pattern Recognition. 2023.
> [2] Zheng, Sipeng, Shizhe Chen, and Qin Jin. "Few-shot action recognition with hierarchical matching and contrastive learning." European Conference on Computer Vision. Cham: Springer Nature Switzerland, 2022.
> [3] Wang, Xiang, et al. "Hybrid relation guided set matching for few-shot action recognition." Proceedings of the IEEE/CVF Conference on Computer Vision and Pattern Recognition. 2022.
>
> **Answer:** In response we provide additional experiments. Table B and C below (Tables 6 and 7 in the appendix) showcase our TCMT$_H$ results against leading metric-based methods like MoLo, HySRM, and HCL in a 5-way-k-shot framework, following your suggestion. In the SSv2-Full and SSv2-Small datasets, we randomly selected 64 classes for $\mathcal{D}$ and 24 for $\mathcal{S}$ and $\mathcal{Q}$. The main difference between SSv2-Full and SSv2-Small is the dataset size, with SSv2-Full containing all samples per category and SSv2-Small including only 100 samples per category. For HMDB-51, we chose 31 action categories for $\mathcal{D}$ and 10 for $\mathcal{S}$ and $\mathcal{Q}$, while for UCF-101, the selection was 70 and 21 categories, respectively. For Kinect, we used 64 action categories for $\mathcal{D}$ and 24 for $\mathcal{S}$ and $\mathcal{Q}$. To maintain statistical significance, we executed 200 trials, each involving random samplings across categories. After training on $\mathcal{D}$, we used $k$ video sequences from each action category to form $\mathcal{S}$ for model updates. The inference phase utilized the remaining data from $\mathcal{Q}$.
>
> **Table B:** Comparing TCMT$_H$ to benchmarks for 5-way-k-shot learning on the SSv2 and SSv2-small
>
> |  |   | SSv2 |  |   | SSv2-small |  |
> |--------|:--------:|:--------:|--------:|--------:|:--------:|--------:|
> |  |  1-shot | 3-shot | 5-shot| 1-shot | 3-shot | 5-shot|
> | OTAM | 42.8 | 51.5 | 52.3 |36.4 | 45.9 | 48.0
> | TRX | 42.0 | 57.6 | 62.6 | 36.0 | 51.9 | 56.7
> | STRM | 42.0 | 59.1| 68.1 | 37.1 | 49.2 | 55.3
> | HyRSM | 54.3 | 65.1 | 69.0 | 40.6 | 52.3 | 56.1
> | HCL | 47.3 | 59.0 | 64.9 | 38.7 | 49.1 | 55.4
> | MoLo | 56.6 | 67.0 | 70.6  | 42.7 | 52.9 | 56.4
> | TCMT$_H$ | **60.0** | **68.3** | **71.9** | **45.8** | **53.6** | **58.0**
>
> **Table C:** Comparing TCMT$_H$ to benchmarks for 5-way-k-shot learning on the UCF-101, HMDB-51, and Kinectics datasets.
>
> |  |  | UCF-101  |   |  HMDB-51 |   | Kinectic|
> |--------|:--------:|:--------:|--------:|--------:|:--------:|--------:|
> |  |  1-shot | 5-shot | 1-shot| 5-shot | 1-shot | 5-shot|
> | OTAM | 79.9 | 88.9  | 54.5 | 68.0 | 79.9 | 88.9
> | TRX | 78.2 | 96.1 | 53.1 | 75.6 | 78.2 | 96.2
> | STRM | 80.5 | **96.9** | 52.3 | 77.3 | 80.5 | 96.9
> | HyRSM | 83.9 | 94.7 | 60.3 | 76.0 | 83.9 | 94.7
> | HCL | 82.8 | 93.3 | 59.1 | 76.3 | 73.7 | 85.8
> | MoLo | 86.0 | 95.5 | 60.8 | 77.4 | 86.0 | 95.5
> | TCMT$_H$ | **87.3** | 96.5 | **61.9** | **80.5** | **86.1** | **98.0**
>
> We observe that TCMT$_H$ had the highest accuracy in 11 out of 12 of these experiments, and had the second highest accuracy in the remaining experiment, trailing by only 0.4.

---

> ### Author Response · Authors · 2023-11-16
> **Response to Reviewer K1pA P3**
>
> >**Weakness 4:** There is no justification whether the causal relationship is learned correctly besides the performance improvement.
>
> **Answer:** Thank you for raising this point. We would like to address your question from the following points.
>
> 1. **No ground-truth:** In our work, the causal relationships are inherently latent, and a significant challenge arises from the lack of ground-truth causal relations in the dataset.
>
> 2. **Alternative measurement:** An possible alternative is that ELBO can help to assess how well the causal representations are captured under the constraint of the identifiability results, which is based on Independence Noise Conditoin (IN). To justify this point, we use ELBO as a metric to conduct an experiment on the Sth-Else dataset. We compare the full $\text{TCMT}_C$ against non-temporal baseline.
> We would like to understand that, if ELBO can measure our causal representation versus the one without temporal causal relationships. Given the importance of temporal causal relationships for sequential data such as videos, this comparison is helpful to understand the impact of these relationships on model performance.
>
> 3. **Preliminary justifications:** We noted that the non-temporal baseline achieved ELBO values of 9.42 for 5-shot and 6.29 for 10-shot settings on the Sth-Else dataset. In comparison, our TCMT model recorded ELBO scores of 7.51 for 5-shot and 4.27 for 10-shot settings. These findings align with the results presented in Table 5 of the main paper, suggesting that lower ELBO scores indicate superior learning of causal representations, which in turn leads to improved recognition performance.
>
> 4. **Future work:** It is noteworthy that since ELBO can capture the causal relationships based on some assumptions like IN, which is challenging to verify in the real-world data. Thus, the best way to measure it is to compare with ground truth.
> One potential method is to utilize synthetic data with ground-truth of the causal relationships. However, currently, there are no suitable synthetic datasets available for few-shot action recognition. Recognizing the importance of this aspect, we plan to address this gap in our future work, potentially through the development or identification of appropriate synthetic datasets that allow for a more direct validation of learned causal relationships. If there is a particular method of verifying the quality of the causal relationship that you would suggest for our setting we would appreciate the recommendation.
>
> >**Weakness 5:** For the comparison number of parameters, all parameters besides the parameters in the adaption process should be counted since they are needed for inference.
>
> **Answer:** Including the backbone, the total number of parameters in our models are 74M for $\text{TCMT}_H$ and 96M for $\text{TCMT}_C$. In comparison, ViFi-CLIP has 124M parameters, VL-Prompting contains 135M, ORViT possesses 148M, and SViT has 152M parameters.
>
> It's important to note that our primary goal is to facilitate efficient adaptation of the model from base to novel data. Therefore, in the body we focus on the number of parameters relevant for this transfer process.
>
> >**Question 0:** It is very slow to open and scroll the submitted document locally. Perhaps Figure 1 (b) has too many objects. I don’t know if this only happen on my site.
>
> **Answer:** This is not unique to you. This is because the images in Figure 1 are populated with a large number of data points, maintaining high resolution.
>
> >**Question 1:** For equation (11), is the ratio of $L_{ELBO}$ and $L_{cls}$ 1:1?
>
> **Answer:** Yes, it is 1:1.
>
> >**Question 2:** Just for curiosity, does the hidden variable theta have interpretable meanings? If theta control certain aspects of the action generation process, it would be easier to justify the causal relationship.
>
> **Answer:** $\theta$ is not easily interpretable because we do not have a ground truth to use for interpretation from the real-world data.
> We probably can understand $\theta$ from a toy example: "Sliding downhill" always involves downward motion with some constant rate of acceleration (possibly zero if speed is constant). At each time step our observation $x$ presents the position of the object while the latent variable $z$ can capture the object's velocity and displacement. The transition function models how the velocity and displacement change over time, while the mixing function outputs the position of the object as a function of its velocity and displacement. The auxiliary variable $\theta$ captures aspects like the angle of motion and acceleration. If we learn "sliding downhill"  on the base data, we should be able to learn “dropping" on the novel data by updating only $\theta$ and the classifier. However,  with the real world data, $\theta$ will not be easily interpreted.
>
> One possible future way to interpreter $\theta$ would be to investigate interventions for post-hoc analysis, i.e., examine the effects of changing $\theta$.

---

> > ### Comment · Reviewer_K1pA · 2023-11-23
> > **Second-round feedback to authors**
> >
> > Thanks the authors for the rebuttal. I am satisfied with the answers to my concerns.

---

> ### Author Response · Authors · 2023-11-16
> **Response to Reviewer K1pA P4**
>
> >**Question 3:** To training the autoencoder, joint training may not be optimal. If the CVAE is firstly trained for causal modeling and then jointly trained for maximizing ELBO and classification, maybe the causal relationship can be better learned. In addition, the results from the first step can be used to verify if the causal relationship is correctly captured.
>
> **Answer:** Excellent point.
> We have added experimental results in Table D below (Table 11 in the appendix) using the Sth-Else dataset to address this point. First we train the autoencoder and context network using ELBO, and then train the classifier separately. There does not appear to be big advantage from training separately.
>
> **Table D:** Ablation study for separate training vs. joint training.
> |  |  5-shot  | 10-shot |
> |--------|:--------:|:--------:|
> | Joint Training  | 48.5 | 59.9 |
> | Separate Training | 48.5 | 60.0 |
>
> However, we agree that it is possible that ELBO is helpful to understand how well the causal representations are learned. To illustrate this, we offer a comparative analysis by comparing the ELBO scores obtained by non-temporal baseline and our TCMT.  The non-temporal baseline achieves ELBO values of 9.42 for 5-shot and 6.29 for 10-shot settings on the Sth-Else dataset. In comparison, our TCMT model recorded ELBO scores of 7.51 for 5-shot and 4.27 for 10-shot settings. These outcomes demonstrate that our TCMT, with its temporal causal relationships, achieves a superior ELBO score.
>
> In light of your suggestion, we are now trying to expand the two-stage training on other datasets. Due to the computation cost of large-scale pre-training, we are not sure whether it can be done before the 22nd Nov. We will update the results at our earliest once we obtain them.
>
> >**Question 4:** In Table 5, what is the “N” used for non-causal, non-temporal, and without theta?
>
> **Answer:** N = 12 for the ablation studies of non-causal, non-temporal, and without $\theta$. We have edited our mention of this at the beginning of paragraph 2 of Section 3.3 to make it more explicit.

---

> ### Author Response · Authors · 2023-11-21
> **Update our response to Question 3**
>
> We have conducted extensive experiments to assess the effectiveness of separate training compared to joint training on the SSv2, HMDB-51, and UCF-101 datasets, using all-way-k-shot settings. The results are meticulously detailed in Tables E and F for general performance, and Tables G and onward focus on the Evidence Lower Bound (ELBO) outcomes for each training method.
>
> Upon analyzing these results, it's clear that the superiority of separate training over joint training is not conclusive. For example, in our study, joint training outperforms separate training in half of the experiments (6 out of 12). Furthermore, the ability of ELBO in separate training to better capture causal relationships than joint training is still uncertain. In our findings, joint training surpasses separate training in one-third of the experiments (4 out of 12), with both methods achieving identical scores in 3 out of 12 experiments.
>
> We recognize the critical importance of accurately measuring the correctness of learned causal relationships, especially in the absence of ground truth data. This remains a complex and challenging issue, and we are committed to exploring it more thoroughly in our future research endeavors, bearing in mind the time-sensitive nature of rebuttal.
>
> **Table E:** Comparing the accuracy of separate training to joint training for all-way-k-shot on the SSv2 dataset
> |  |  | SSv2  |   |   |
> |--------|:--------:|:--------:|--------:|--------:|
> |  |  2-shot | 4-shot | 8-shot| 16-shot |
> | Joint training |  7.5 | 9.6 |11.8 |15.5
> | Separate training | 8.1 | 9.5 | 11.4 |16.1
>
> **Table F:** Comparing the accuracy of separate training to joint training for all-way-k-shot on the HMDB-51 and UCF-101 datasets
> |  |  |  | HMDB-51   |   |   |  UCF-101  |   |   |
> |--------|:--------:|:--------:|--------:|--------:|:--------:|:--------:|--------:|--------:|
> |  |  2-shot | 4-shot | 8-shot| 16-shot |  2-shot | 4-shot | 8-shot| 16-shot |
> | Joint training |  65.8 | 70.2 | 72.5 | 75.7 |  90.6 | 94.7 | 96.2 | 98.5
> | Separate training | 65.5 | 70.8 | 72.2 | 75.9 | 89.7 | 94.9 | 96.1 | 98.4
>
>
> **Table G:** Comparing the ELBO of separate training to the ELBO of joint training on the Sth-Else dataset
> |  |  5-shot  | 10-shot |
> |--------|:--------:|:--------:|
> | Joint Training  | 6.29 | 4.27 |
> | Separate Training | 6.15 | 4.27 |
>
>
> **Table H:** Comparing the ELBO of separate training to the ELBO of joint training for all-way-k-shot on the SSv2 dataset
> |  |  | SSv2  |   |   |
> |--------|:--------:|:--------:|--------:|--------:|
> |  |  2-shot | 4-shot | 8-shot| 16-shot |
> | Joint training |  36.8 | 34.0 | 31.4 | 29.1
> | Separate training | 36.5 | 34.0 | 31.9 | 29.1
>
>
> **Table I:** Comparing the ELBO of separate training to the ELBO of joint training for all-way-k-shot on the HMDB-51 and UCF-101 datasets
> |  |  |  | HMDB-51   |   |   |  UCF-101  |   |   |
> |--------|:--------:|:--------:|--------:|--------:|:--------:|:--------:|--------:|--------:|
> |  |  2-shot | 4-shot | 8-shot| 16-shot |  2-shot | 4-shot | 8-shot| 16-shot |
> | Joint training |  4.66 | 4.50 | 4.18 | 4.11 | 2.56 | 2.39 | 2.37 | 2.26
> | Separate training | 4.66 | 4.47 | 4.22 | 4.09 | 2.57 | 2.41 | 2.37 | 2.26

---

> ### Author Response · Authors · 2023-11-21
> **Have your concerns been properly addressed?**
>
> Dear Reviewer K1pA,
>
> Thank you for the time and effort you have dedicated to reviewing and providing feedback on our submission. Hopefully our responses and the revisions made to our work effectively address your concerns. Should there be any additional points or matters you wish for us to consider, we are keen and ready to respond to them.
>
> Authors of submission 342

---

> ### Author Response · Authors · 2023-11-23
> **Could you update the score and confidence?**
>
> Dear Reviewer K1pA,
>
> Thank you for your response and for acknowledging our efforts in addressing your comments. We would greatly appreciate it if you could consider updating your evaluation score and confidence level. Should you require any further clarifications, we are fully committed to addressing them to the best of our ability for the remainder of discussion period.

---

### Official Review · Reviewer_sFNL · 2023-10-31

**Soundness:** 2 fair
**Presentation:** 1 poor
**Contribution:** 3 good
**Rating:** 5
**Confidence:** 3

**Summary:**

This paper proposes a method for solving few-shot action recognition, which utilises the idea of variational inference to solve the problem, effectively reducing the number of parameters to be learned during adapation phase.

**Strengths:**

Pros:
1. The basic motivation is feasible.
2. The paper gives a good theoretical analysis.

**Weaknesses:**

Cons:
1. The paper mentions that TCMT is capable of “adapt a base model effectively and efficiently when the base and novel data have significant distributional disparities.” However, there is no experimental verification of such performance, and it is hoped that additional experiments in this area or further additions will be made to show that the existing dataset satisfies such conditions.
2. The authors should add an experiment on the observed time frequency to the section on ablation experiments.
3. This paper needs further improvement in the writing. For example, in Fig.2, $Z_{1,1}$  has an extra bracket around the variable. And all tables in the paper should be of a uniform size. There are numerous other grammatical errors that I have not mentioned but which take away from the reading experience significantly. I hope the author will review and correct these.

**Questions:**

As mentioned above, how does TCMT perform when the base and novel data have significant distributional disparities?

---

> ### Author Response · Authors · 2023-11-16
> **Response to Reviewer sFNL**
>
> We thank the reviewer for the time and comments on our paper. The following is our response.
>
> >**Weakness 1:** The paper mentions that TCMT is capable of “adapt a base model effectively and efficiently when the base and novel data have significant distributional disparities.” However, there is no experimental verification of such performance, and it is hoped that additional experiments in this area or further additions will be made to show that the existing dataset satisfies such conditions.
>
> **Answer:**
> We have justified the capability of TCMT to handle the distributional disparities from following two aspects:
>
> **Comparing with the State-Of-The-Art:** *In our initial submission*,  Figure 1 (a)  in our paper contains a UMap visualization showing the severity of distributional disparities between base and novel data on the Sth-Else dataset. For effectiveness, Table 1 demonstrates the advantage of our TCMT against other state-of-the-art methods, and our ablation studies verify that this benefit comes from handling these distributional disparities.
>
> **The efficacy of causal representation:** We have now extended our ablation study with Table A below (it has been added as Table 8 in the appendix) to show that TCMT outperforms other methods even when they augmented with auxiliary context variables, thus validating the important of the causal approach. Figure 4 demonstrates the improvement in efficiency during adaptation by comparing the number of parameters that need to be updated using the novel data.
>
> **Table A:** Additional comparisons by augmenting existing methods on the Sth-Else dataset. $+\theta$ means the method updates the Context Network when adapting instead of fine-tuning. Since VL Prompting uses VPT \citep{vpt_eccv22} within the ViFi-CLIP framework, we only test ViFi-CLIP$+\theta$.
> | Method |  5-shot | 10-shot |
> |--------|:--------:|:--------:|
> | ORViT | 33.3 | 40.2 |
> | ORViT+$\theta$ | 33.9 | 41.8 |
> | SViT | 34.4 | 42.6 |
> | SViT+$\theta$ | 35.2| 44.0 |
> | TCMT$_H$ | 37.6| 44.0 |
> | ViFi-CLIP |44.5| 54.0 |
> | ViFi-CLIP+$\theta$ |45.2| 58.0 |
> | VL Prompting |44.9| 58.2 |
> |TCMT$_C$ |**48.5**| **59.9** |
>
> To further validate fixing the transition and mixing functions during adaptation, we conduct additional comparisons between TCMT$_C$ and a fine-tuning baseline, TCMT-FT, with the all-way-k-shot settings. The results, as presented in Table B and Table C below  (Table 5, as well as Tables 9 and 10 in our submission), confirm that TCMT$_C$ generally achieves better scores compared to TCMT-FT.
>
> **Table B:** Comparing TCMTC to TCMT-FT for all-way-k-shot on the SSv2 dataset
> |  |  | SSv2  |   |   |
> |--------|:--------:|:--------:|--------:|--------:|
> |  |  2-shot | 4-shot | 8-shot| 16-shot |
> | TCMT-FT | 6.1 | 7.9 | 10.4 |14.1
> | TCMT$_C$ |  7.5 | 9.6 |11.8 |15.5
>
> **Table C:** Comparing TCMTC to TCMT-FT for all-way-k-shot on the HMDB-51 and UCF-101 datasets
> |  |  |  | HMDB-51   |   |   |  UCF-101  |   |   |
> |--------|:--------:|:--------:|--------:|--------:|:--------:|:--------:|--------:|--------:|
> |  |  2-shot | 4-shot | 8-shot| 16-shot |  2-shot | 4-shot | 8-shot| 16-shot |
> | TCMT-FT | 61.5 | 64.8 | 70.0 | 72.9 | 87.7 | 93.7 | 95.1 | 96.4
> | TCMT$_C$ |  65.8 | 70.2 | 72.5 | 75.7 |  90.6 | 94.7 | 96.2 | 98.5
>
> >**Weakness 2:** The authors should add an experiment on the observed time frequency to the section on ablation experiments.
>
> **Answer:** Thank you for drawing our attention to this.
> We have rerun our ablation experiment on the Sth-Else dataset with 4, 8 and 16 of frames per video sequence uniformly spaced to see if this has any impact on the results. The results are reported in the Table D, which is Table 9 in the appendix. When we use 16 frames as the input, we observe a slight improvement. However using 16 frames doubles the computational cost, requiring eight 2080-ti GPUs to complete training within 24 hours.
>
> **Table D:** Ablation study for selecting the different length of input of TCMT$_C$ on the Sth-Else dataset..
> | Input frames |  5-shot | 10-shot |
> |--------|:--------:|:--------:|
> | 4 | 43.3 | 51.0 |
> | 8 | 48.5 | 59.9 |
> | 16 | 50.8 | 61.2 |
>
> >**Weakness 3:** This paper needs further improvement in the writing. For example, in Fig.2, has an extra bracket around the variable. And all tables in the paper should be of a uniform size. There are numerous other grammatical errors that I have not mentioned but which take away from the reading experience significantly. I hope the author will review and correct these.
>
> **Answer:** We have fixed the typo in Figure 2 and all other typos that we have found. We are happy to edit any other typographical or grammatical errors that are brought to our attention. However, we do not find it beneficial to make all tables a uniform size, and this is not standard in ICLR papers. This would require small tables to take up large amounts of space, for large tables to be squeezed too small to be readable, or for larger tables to be broken up.

---

> > ### Author Response · Authors · 2023-11-21
> > **Any further comments?**
> >
> > Dear Review sFNL
> >
> > Thank you again for dedicating your time and effort to reviewing our work. If you have any further questions or points for us to address, please do not hesitate to let us know. Given the time-sensitive nature of this rebuttal process, we greatly appreciate your prompt response.
> >
> > The authors of 342

---

> > ### Comment · Reviewer_sFNL · 2023-11-22
> >
> > Thank you for your careful answer to my confusion. But I have some confusion as below, regarding Weakness 1, whether such phenomenon is commonly found in different datasets and how such distributional disparities affect the performance of the model. (It might be more intuitive to give some sample examples here, the brackets are only suggestions and will not affect the final scoring)
> >
> > Similarly, I'm concerned about the validity of the assumptions made by other reviewers about the assumptions in INTRODUCTION. You can respond directly to other reviewers' related confusions, and I'll measure my score directly against your responses under their reviews.

---

> ### Author Response · Authors · 2023-11-22
> **Response to the additional comment**
>
> Thank you for your additional comment.
>
> We assume "such phenomenon" to refer to the distributional disparities observed in the datasets used for our experiments and how our Temporal Causal Mechanism Transfer (TCMT) model addresses these disparities. Figures 7, 8, and 9 in the appendix of our revised submission illustrate these distributional disparities between base and novel data under our all-way-k-shot settings. These figures also support our assumption that the transition and mixing functions remain invariant across the base and novel data.
>
> It is a common assumption, as highlighted by our comparing approaches [1, 2], that the base and novel data distributions differ. Generally, in few-shot learning tasks, a larger disparity between these distributions makes it more challenging for models to adapt to novel data. For example, the authors of [3] explicitly mention that a method can "fail to generalize to unseen domains due to a large discrepancy in feature distribution." A similar observation is made by the authors of [4]. This aligns with our experimental results. For instance, Figure 7 in our revision's appendix shows that the distributions from the base dataset (K-400) and the novel dataset (UCF-101) have a large discrepancy, leading to low accuracy scores reported in Table 3. However, the superior scores achieved by TCMT underscore the better effectiveness of TCMT comparing with other approaches.
>
> Regarding "the assumptions made by other reviewers," we understand you are referencing Question 2 posed by Reviewer G3yX. We had provided a response to this query BEFORE receiving your further comment. In light of your suggestions, we will update our response to Reviewer G3yX as well.
>
> Additionally, *we note that the concerns you raised, particularly those pertaining to specific points in brackets, appear to have significantly influenced your scoring of our representations.* We would greatly appreciate any further clarifications you could provide on these issues.
>
> [1] Phoo, Cheng Perng, and Bharath Hariharan. "Self-training For Few-shot Transfer Across Extreme Task Differences." International Conference on Learning Representations. 2021.
>
> [2] Wang, Xiang, et al. "Cross-domain few-shot action recognition with unlabeled videos." Computer Vision and Image Understanding (2023): 103737.
>
> [3] Tseng, Hung-Yu, et al. "Cross-Domain Few-Shot Classification via Learned Feature-Wise Transformation." International Conference on Learning Representations. 2020.
>
> [4] Luo, Xu, et al. "A Closer Look at Few-shot Classification Again." arXiv preprint arXiv:2301.12246 (2023).

---

### Official Review · Reviewer_G3yX · 2023-11-01

**Soundness:** 2 fair
**Presentation:** 2 fair
**Contribution:** 3 good
**Rating:** 5
**Confidence:** 2

**Summary:**

This paper proposes a few-shot learning for action recognition based on temporal casual representation, called Temporal Causal Mechanism Transfer. The method is built on an assumption that the base data and novel data share certain aspects of the temporal causal mechanism, transition function and mixing function. It conducts experiments on multiple datasets and achieves great performance. Thw writing is somehow good.

**Strengths:**

1. The idea of using temporal causal mechanism for few-shot video recognition is new to me.
2. The method is effective and achieves good results on multiple datasets.

**Weaknesses:**

1. The third paragraph in the Intro is very highlight and intuitive. The motivation of using casual representation for few-shot action recognition is not clear to me from the paper.
2. Fig. 2 lacks illustration in both caption and main contents. I can not understand well the methods without much casual representation background. And there is less introduction for the causal representation.
3. All datasets miss details.
4. Miss conclusions for all figures of results. The statements for results only list numbers but lack analysis. For example, in Fig. 5, the paper compares the proposed method and a previous method VL-Prompting. What's the difference between the two methods? What makes difference between their results? Why the proposed one is better than the previous one?

**Questions:**

I have two very serious question. Without clarification on the two points, I can not understand the paper well.

1. What’s the motivation/intuition to use causal representation learning for few-shot action recognition? I feel it is not clear to me from the paper.
2. In the third paragraph in Intro, there is an assumption "the base data and novel data share certain
aspects of the temporal causal mechanism – namely, transition function and mixing function – and
that an auxiliary variable captures the disparate aspects of the two data distributions" which is the base of the method. However, I can not find why the assumption is acceptable?  Is there any support or reference?

---

> ### Author Response · Authors · 2023-11-16
> **Response to Reviewer G3yX P1**
>
> We are grateful for the time you spent on our paper and suggestions, the following is our response.
>
> >**Weakness 1:** The third paragraph in the Intro is very highlight and intuitive. The motivation of using casual representation for few-shot action recognition is not clear to me from the paper.
>
> **Answer:**
> To make few-shot learning more efficient and effective, we would like to train a model such that parts of the model can remain fixed during adaptation. The fewer parameters we have to update, the more efficient adaptation will be.
> *We therefore need to understand what causes the distributional disparities, which is unanswered in the current literature on few-shot action recognition. This motivates us to model a causal mechanism to identify what components of the model can be held fixed and what factors lead to disparities in the action representations and labels.* The literature on causal representation learning and principle of ``minimum change" give us the theoretical grounding to isolate the factors that cause disparities so that we can hold those parts of our model fixed during adaptation [1,2].
>
> By developing a temporal causal mechanism we are able to separate (1) the causal relationships between latent causal variables over time (transition function), (2) the dependence of the observation on the causal variables at a given time step (mixing function), and other time-dependent causal factors that are likely to vary between the base and novel data (auxiliary variable).
>
> >**Weakness 2:** Fig. 2 lacks illustration in both caption and main contents. I can not understand well the methods without much casual representation background. And there is less introduction for the causal representation.
>
> **Answer:** Consider the following toy example for intuition. "Sliding downhill" always involves downward motion with some constant rate of acceleration (possibly zero if speed is constant). At each time step our observation $x$ presents the position of the object while the latent variable $z$ can capture the object's velocity and displacement. The transition function models how the velocity and displacement change over time, while the mixing function outputs the position of the object as a function of its velocity and displacement. The auxiliary variable $\theta$ captures aspects like the angle of motion and acceleration. If we learn "sliding downhill"  on the base data, we should be able to learn "dropping" on the novel data by updating only $\theta$ and the classifier.
>
> The contents of Figure 2 (a) (Figure 2 in original submission) are discussed in detail in the fourth paragraph of the introduction. The purpose of the figure is only to add intuition by illustrating the in-depth descriptions which directly precede it. We choose not extend the caption because the figure environment is narrow, which would lead the caption to be very long, and the caption contents would be highly redundant.
>
> We believe that a comprehensive background on causal representation learning would be too distracting to our paper, and take up considerable space. We believe the ``Generative Model" section of our methodology should be sufficient for understanding  the aspects of causal representation learning relevant to our work and we give the critical references for related work on causal representation (the third paragraph in the introduction and the "causal representation learning" part in the related work). Appendix A.1 gives further details on the role of the independent noise condition used for causal representation learning.
>
> >**Weakness 3:** All datasets miss details.
>
> **Answer:** Thank you for drawing our attention to this.
> We realize that we have provided the references in the paper for describing the dataset, but the details should be added for clarification. We have added them in Appendix A.2:
>
> 1. Something-Something v2 (SSv2) is a dataset containing 174 action categories of common human-object interactions;
>
> 2. Something-Else (Sth-Else) exploits the compositional structure of SSv2, where a combination of a verb and a noun defines an action;
>
> 3. HMDB-51 contains 7k videos of 51 categories;
>
> 4. UCF-101 covers 13k videos spanning 101 categories;
>
> 5. Kinetics covers around 230k 10-second video clips sourced from YouTube.
>
> Please refer to the ``Datasets" section in the main paper for the segmentation of $\mathcal{D}$, $\mathcal{S}$ and $\mathcal{Q}$.

---

> > ### Comment · Reviewer_G3yX · 2023-11-30
> > **For weakness 2**
> >
> > Feel good for the answers to weakness 1 and 3.
> > For weakness 2, clear paper background is very important for me to understand the paper. At least some background knowledges should be introduced.

---

> ### Author Response · Authors · 2023-11-16
> **Response to Reviewer G3yX P2**
>
> >**Weakness 4:** Miss conclusions for all figures of results. The statements for results only list numbers but lack analysis. For example, in Fig. 5, the paper compares the proposed method and a previous method VL-Prompting. What's the difference between the two methods? What makes difference between their results? Why the proposed one is better than the previous one?
>
> **Answer:** For all comparisons we make to all other models, ours is the only causal model based on causal representation. VL-Prompting does not invoke a causal model and therefore struggles to handle certain distribution disparities between the base and novel data. VL-Prompting is also shown in the plot of Figure 4, demonstrating that it requires a far greater number of parameters to be updated during transfer.
>
> We would like to draw attention to the results in Table 1 of our paper which demonstrate the superiority of our TCMT framework over state-of-the-art methods like ViFi-CLIP and VL-Prompting. We succinctly describe both ViFi-CLIP and VL-Prompting in the second paragraph in the introduction, and third paragraph of the Related Work section, highlighting that while these methods successfully leverage the pre-trained CLIP model for few-shot action recognition, they may fall short when faced with significant distribution disparities between base and novel datasets. TCMT is distinguished by its use of causal representation learning to construct generative models of temporal causal processes, a strategy that our results show is particularly effective in addressing distribution disparities. This conclusion is not only theoretically sound but is also empirically validated through our experiments, underscoring TCMT's superiority over methods like VL Prompting.
>
> For the statements of results, it is not entirely clear to us what you mean when you say we lack analysis, and clarification would be greatly appreciated. It would greatly benefit us if you could be more explicit with the form of analysis you have in mind. We include explanations for each of our experiments, tables, and figures.
>
> >**Question 1:** What’s the motivation/intuition to use causal representation learning for few-shot action recognition? I feel it is not clear to me from the paper.
>
> **Answer:** Consider the following toy example for intuition. "Sliding downhill" always involves downward motion with some constant rate of acceleration (possibly zero if speed is constant). At each time step our observation $x$ presents the position of the object while the latent variable $z$ can capture the object's velocity and displacement. The transition function models how the velocity and displacement change over time, while the mixing function outputs the position of the object as a function of its velocity and displacement. The auxiliary variable $\theta$ captures aspects like the angle of motion and acceleration. If we learn "sliding downhill"  on the base data, we should be able to learn "dropping" on the novel data by updating only $\theta$ and the classifier.
>
> To make few-shot learning more efficient and effective, we would like to train a model such that parts of the model can remain fixed during adaptation. The fewer parameters we have to update, the more efficient adaptation will be.
> *We therefore need to understand what causes the distributional disparities, which is unanswered in the current literature on few-shot action recognition. This motivates us to model a causal mechanism to identify what components of the model can be held fixed and what factors lead to disparities in the action representations and labels.* The literature on causal representation learning and principle of ``minimum change" give us the theoretical grounding to isolate the factors that cause disparities so that we can hold those parts of our model fixed during adaptation [1,2].
>
> >**Question 2:** In the third paragraph in Intro, there is an assumption "" which is the base of the method. However, I can not find why the assumption is acceptable? Is there any support or reference?
>
> **Answer:** We have justified this assumption in the *initial submission* by:
>
> 1. Figure 1b gives a snapshot of the invariance exhibited by the transition function between the base and novel data of the Sth-Else dataset. We have added a figure to show the invariance captured by the mixing function (Figure 1 (c) in the updated paper). We therefore expect limited benefit from updating these components during adaptation, and holding them fixed acts as a form of regularization to prevent over-fitting.
>
> 2. Our original Figure 1 (c) (Figure 2 (b) in the updated version) shows that when holding the transition and mixing function fixed we see faster convergence to a higher accuracy on the Sth-Else dataset compared to updating the transition and mixing functions. We therefore have strong motivation to hypothesize that updating fewer parameters both improves efficiency and acts as a form of regularization. Our main experiments bear this out.

---

> > ### Author Response · Authors · 2023-11-16
> > **Response to Reviewer G3yX P3**
> >
> > [1] Kong, Lingjing, et al. "Partial disentanglement for domain adaptation." International Conference on Machine Learning. PMLR, 2022.
> >
> > [2] Xie, Shaoan, et al. "Multi-domain image generation and translation with identifiability guarantees." The Eleventh International Conference on Learning Representations. 2023.

---

> > ### Author Response · Authors · 2023-11-18
> > **Update our answer for Question 2**
> >
> > To further validate fixing the transition and mixing functions during adaptation, we conduct additional comparisons between TCMT$_C$ and a fine-tuning baseline, TCMT-FT, across all datasets in the all-way-k-shot settings. The results, as presented in Table A and Table B below  (Table 5, as well as Tables 9 and 10 in our submission), confirm that TCMT$_C$ generally achieves better scores compared to TCMT-FT.
> >
> > **Table A:** Comparing TCMTC to TCMT-FT for all-way-k-shot on the SSv2 dataset
> > |  |  | SSv2  |   |   |
> > |--------|:--------:|:--------:|--------:|--------:|
> > |  |  2-shot | 4-shot | 8-shot| 16-shot |
> > | TCMT-FT | 6.1 | 7.9 | 10.4 |14.1
> > | TCMT$_C$ |  7.5 | 9.6 |11.8 |15.5
> >
> > **Table B:** Comparing TCMTC to TCMT-FT for all-way-k-shot on the HMDB-51 and UCF-101 datasets
> > |  |  |  | HMDB-51   |   |   |  UCF-101  |   |   |
> > |--------|:--------:|:--------:|--------:|--------:|:--------:|:--------:|--------:|--------:|
> > |  |  2-shot | 4-shot | 8-shot| 16-shot |  2-shot | 4-shot | 8-shot| 16-shot |
> > | TCMT-FT | 61.5 | 64.8 | 70.0 | 72.9 | 87.7 | 93.7 | 95.1 | 96.4
> > | TCMT$_C$ |  65.8 | 70.2 | 72.5 | 75.7 |  90.6 | 94.7 | 96.2 | 98.5

---

> > > ### Author Response · Authors · 2023-11-22
> > > **Adding figures to update our answer to question 2**
> > >
> > > To enhance our response to Question 2 and offer a more comprehensive explanation, we have included Figures 7, 8, and 9 in the appendix of our revised submission. These figures clearly depict the distributional disparities between base and novel data across all datasets under our all-way-k-shot settings. Importantly, they also corroborate our assumption that both the transition and mixing functions maintain invariance across the base and novel data sets. This addition is intended to provide a clearer understanding and stronger support for our methodology and findings.

---

> > ### Comment · Reviewer_G3yX · 2023-11-30
> > **Weakness 4**
> >
> > As I mentioned in the weakness, only result numbers are shown in the paper. There are no reason analysis for why the proposed is better than others.

---

> ### Author Response · Authors · 2023-11-21
> **Do the answers address your concerns?**
>
> Dear Reviewer G3yX,
>
> Thank you for your time and comments. Could you let us know your thoughts on whether our responses and revisions have adequately addressed your concerns? We are fully prepared to address any further questions you may have. Your prompt feedback is highly appreciated given we hope to have the opportunity to respond further during the period of discussions.
>
> The author of 342

---

### Official Review · Reviewer_oCST · 2023-11-01

**Soundness:** 3 good
**Presentation:** 2 fair
**Contribution:** 3 good
**Rating:** 5
**Confidence:** 3

**Summary:**

This paper proposes Temporal Causal Mechanism Transfer (TCMT), a new method for few-shot action recognition in videos. The key ideas and contributions are:

- TCMT learns a generative model of a temporal causal process from the base data. This includes a transition function that models time-delayed causal relations between latent variables, and a mixing function that generates action representations from the latent variables.

- For adaptation on novel data, TCMT updates an auxiliary context variable that captures distribution shifts between base and novel data, along with the classifier weights. The transition and mixing functions remain fixed.

- TCMT is evaluated on standard few-shot action recognition benchmarks and achieves state-of-the-art or comparable accuracy with fewer parameter updates during adaptation.

- The effectiveness of TCMT is attributed to the transferability of the learned causal mechanism. Ablations validate the benefits of modeling temporal relations and using auxiliary variables.

- The approach demonstrates the promise of causal representation learning for few-shot action recognition. Limitations include assumptions on temporal delays and difficulty inferring the auxiliary variables.

In summary, the key contribution is a new few-shot learning method based on learning and transferring temporal causal mechanisms, which is shown to be accurate and efficient for adapting models to new action recognition tasks with limited labeled video data.

**Strengths:**

1. Originality: The idea of learning and transferring a temporal causal mechanism is highly original. Causal representation learning has not been applied in this way for few-shot action recognition before. Modeling latent causal variables, time-delayed transitions, and mixing functions is creative.

2. Quality: The method is technically sound, with reasonable assumptions justified from first principles of causality. Experiments across multiple datasets demonstrate state-of-the-art accuracy and efficiency. The ablation study provides insight into design choices.

3. Clarity: Overall the paper is clearly written and easy to follow. The background gives sufficient context, and the methodology explains the approach in detail. More intuition could be provided for how the causal mechanism aids adaptation.

4. Significance: This provides a new paradigm for few-shot video understanding based on causal representation learning. The ability to adapt models with fewer updates could enable deploying action recognition systems to new domains with limited labeled data. Limitations around temporal delays and auxiliary variables indicate interesting directions for future work.

**Weaknesses:**

1. The motivation for why the causal mechanism transfers well could be clarified. Intuition or analysis on how the transition and mixing functions capture invariances would strengthen the core hypothesis.
2. The inference of the auxiliary context variables θ seems coarse. More details on this convolutional LSTM approach and why it is effective would be helpful. Alternate ways to model θ could improve performance.
3. Assumptions like time-delayed transitions between latent variables may not hold for data with low time resolution. Discussion of this limitation and ways to incorporate instantaneous effects would make the model more broadly applicable.
4. More comparisons to understand tradeoffs versus other representation learning approaches like self-supervision may be informative.

**Questions:**

Please see the 'weaknesses' above.

---

> ### Author Response · Authors · 2023-11-16
> **Response to Reviewer oCST P1**
>
> We sincerely appreciate your efforts and suggestions on our paper and provide our feedback below:
>
> >**Weakness 1:** The motivation for why the causal mechanism transfers well could be clarified. Intuition or analysis on how the transition and mixing functions capture invariances would strengthen the core hypothesis.
>
> **Answer:** Our answer is as following.
>
> **Explanation from a toy example** Consider the following toy example for intuition. "Sliding downhill" always involves downward motion with some constant rate of acceleration (possibly zero if speed is constant). At each time step our observation $x$ presents the position of the object while the latent variable $z$ can capture the object's velocity and displacement. The transition function models how the velocity and displacement change over time, while the mixing function outputs the position of the object as a function of its velocity and displacement. The auxiliary variable $\theta$ captures aspects like the angle of motion and acceleration. If we learn "sliding downhill"  on the base data, we should be able to learn “dropping" on the novel data by updating only $\theta$ and the classifier.
>
> **Motivation** To make few-shot learning more efficient and effective, we would like to train a model such that parts of the model can remain fixed during adaptation. The fewer parameters we have to update, the more efficient adaptation will be.
> *We therefore need to understand what causes the distributional disparities, which is unanswered in the current literature on few-shot action recognition. This motivates us to model a causal mechanism to identify what components of the model can be held fixed and what factors lead to disparities in the action representations and labels.* The literature on causal representation learning and principle of ``minimum change" give us the theoretical grounding to isolate the factors that cause disparities so that we can hold those parts of our model fixed during adaptation [1,2].
>
> By developing a temporal causal mechanism we are able to separate (1) the causal relationships between latent causal variables over time (transition function), (2) the dependence of the observation on the causal variables at a given time step (mixing function), and other time-dependent causal factors that are likely to vary between the base and novel data (auxiliary variable).
>
> **Justification** In our initial submission, Figure 1b gives a snapshot of the invariance exhibited by the transition function between the base and novel data of the Sth-Else dataset. We have now added a UMap figure to show the invariance captured by the mixing function as well in Figure 1 (c) of the updated paper. We therefore expect limited benefit from updating these components of our model during adaptation, and holding them fixed acts as a form of regularization to prevent over-fitting. Our Figure 2 (b) (Figure 1 (c) from the initial submission) shows that when holding the transition and mixing function fixed we see faster convergence to a higher accuracy on the Sth-Else dataset compared to updating the transition and mixing functions. We therefore have strong motivation to hypothesize that updating fewer parameters both improves efficiency and acts as a form of regularization. Our main experiments bear this out.
>
> >**Weakness 2:** The inference of the auxiliary context variables $\theta$ seems coarse. More details on this convolutional LSTM approach and why it is effective would be helpful. Alternate ways to model $\theta$ could improve performance.
>
> **Answer:** *This is a limitation we identified explicitly in our conclusions and motivates future work.* Recent theoretical work [1,2] has demonstrated identifiability results built upon $\theta$. To interpret $\theta$ at a ``fine-grained" level, a deeper understanding of its role in the latent causal process is necessary. This is beyond the limits of the identifiability results in the existing literature. In our work, we experimented with various methods to model $\theta={\theta_1,\theta_2,...,\theta_t}$ as a sequence, including ConvLSTM, 1D CNN, and Transformer. Ultimately, we opted for ConvLSTM based on a balanced consideration of performance and the efficiency of adaptation. As highlighted in our conclusion, this is an area we are keen to explore in our future research.

---

> > ### Author Response · Authors · 2023-11-22
> > **Update our response to Weakness 1**
> >
> > To enhance our justifications to Weakness 1 and offer a more comprehensive explanation, we have included Figures 7, 8, and 9 in the appendix of our revised submission. These figures clearly depict the distributional disparities between base and novel data across all datasets under our all-way-k-shot settings. Importantly, they also corroborate our assumption that both the transition and mixing functions maintain invariance across the base and novel data sets. This addition is intended to provide a clearer understanding and stronger support for our methodology and findings.
> >
> > Moreover, we provide further experimental results to further validate fixing the transition and mixing functions during adaptation, we conduct additional comparisons between TCMT$_C$ and a fine-tuning baseline, TCMT-FT, across all datasets in the all-way-k-shot settings. The results, as presented in Table A and Table B below  (Table 5, as well as Tables 9 and 10 in our submission), confirm that TCMT$_C$ generally achieves better scores compared to TCMT-FT.
> >
> > **Table A:** Comparing TCMTC to TCMT-FT for all-way-k-shot on the SSv2 dataset
> > |  |  | SSv2  |   |   |
> > |--------|:--------:|:--------:|--------:|--------:|
> > |  |  2-shot | 4-shot | 8-shot| 16-shot |
> > | TCMT-FT | 6.1 | 7.9 | 10.4 |14.1
> > | TCMT$_C$ |  7.5 | 9.6 |11.8 |15.5
> >
> > **Table B:** Comparing TCMTC to TCMT-FT for all-way-k-shot on the HMDB-51 and UCF-101 datasets
> > |  |  |  | HMDB-51   |   |   |  UCF-101  |   |   |
> > |--------|:--------:|:--------:|--------:|--------:|:--------:|:--------:|--------:|--------:|
> > |  |  2-shot | 4-shot | 8-shot| 16-shot |  2-shot | 4-shot | 8-shot| 16-shot |
> > | TCMT-FT | 61.5 | 64.8 | 70.0 | 72.9 | 87.7 | 93.7 | 95.1 | 96.4
> > | TCMT$_C$ |  65.8 | 70.2 | 72.5 | 75.7 |  90.6 | 94.7 | 96.2 | 98.5

---

> ### Author Response · Authors · 2023-11-16
> **Response to Reviewer oCST P2**
>
> >**Weakness 3:** Assumptions like time-delayed transitions between latent variables may not hold for data with low time resolution. Discussion of this limitation and ways to incorporate instantaneous effects would make the model more broadly applicable.
>
> **Answer:** *This is another limitation we identified explicitly in our conclusions and motivates future work.* The problem of incorporating instantaneous effects is a substantial problem on extending identifiability and is beyond the scope of our present work. For example, one would need to address cyclic relationships between causal variables, and the available theoretical identifiability results on causal representation learning do not give us a principled approach for doing this.
>
> >**Weakness 4:** More comparisons to understand trade-offs versus other representation learning approaches like self-supervision may be informative.
>
> **Answer:** In the absence of a ground truth for the latent variable $z$, our TCMT approach effectively adopts a self-supervised methodology, utilizing a CVAE. This choice aligns with the requirements for the identifiability of causal representations, as elucidated in [1,2,3] Extending identifiability to other ``self-supervised" frameworks is a valuable and insightful suggestion. We recognize the potential in this direction and will definitely consider it in our future research endeavors. Additionally, we note that we provide direct comparisons with several self-supervised approaches such as SEEN [4] and STARTUP [5] in Table 4. We believe these results clearly demonstrate the advantage of TCMT. If there are particular results from the latest literature on self-supervision for few-shot action recognition which you believe we should add, we are happy to add such comparisons to our paper.
>
> [1] Kong, Lingjing, et al. "Partial disentanglement for domain adaptation." International Conference on Machine Learning. PMLR, 2022.
>
> [2] Xie, Shaoan, et al. "Multi-domain image generation and translation with identifiability guarantees." The Eleventh International Conference on Learning Representations. 2023.
>
> [3] Huang, Biwei, et al. "AdaRL: What, Where, and How to Adapt in Transfer Reinforcement Learning." International Conference on Learning Representations. 2022.
>
> [4] Wang, Xiang, et al. "Cross-domain few-shot action recognition with unlabeled videos." Computer Vision and Image Understanding (2023): 103737.
>
> [5] Phoo, Cheng Perng, and Bharath Hariharan. "Self-training For Few-shot Transfer Across Extreme Task Differences." International Conference on Learning Representations. 2021.

---

> ### Author Response · Authors · 2023-11-21
> **Any further comments?**
>
> Dear Reviewer oCST,
>
> We would like to express our gratitude once again for your time. We hope that our responses have satisfactorily addressed your concerns. If you have any additional comments or questions, we kindly request you to share them with us as soon as possible, so we can provide our responses within the allocated discussion period.
>
> The authors of 342

---

### Author Response · Authors · 2023-11-18
**Summary of the revision**

Dear reviewers,
We express our gratitude for your valuable feedbacks. It is heartening to note that all reviewers acknowledge the contribution of TCMT. We are particularly encouraged *by your recognition of the motivation (Reviewer sNFL and K1pA), novelty (Reviewer oCST, G3yX, and K1pA) and the good results (Reviewer oCST, G3yX, and K1pA) of our work.* In response to your detailed comments, we have made comprehensive revisions, which are summarized in the revision of our paper as follows:

- We clarify our motivation for using causal representation in the third paragraph of introduction (Reviewer oCST and G3yX). Also to better justify our assumption, we add a figure for mixing function capturing the invariance in Figure 1 (c) *(Reviewers oCST, G3yX and sFNL)*. Additionally, we have included Figures 7 to 9 in the Appendix to illustrate the distributional disparities between the base and novel data. These figures also highlight our findings that the transition and mixing functions remain invariant across all datasets used in our all-way-k-shot experiments.


- We add results comparing TCMT and TCMT-FT across all datasets under all-way-k-shot settings. The findings presented in Table 5 of the main paper, as well as Tables 9 and 10 in the Appendix *(Reviewer G3yX and sFNL)*. We also add greater detail about our datasets in A.2 of Appendix *(Reviewer G3yX)*. An experiment on the observed time frequency to the section on ablation experiments is added in Table 11 of Appendix *(Reviewer sFNL)*;

- We add quantitative comparisons between TCMT and state-of-the-art approaches augmented $\theta$ to demonstrate the advantage of our method using causal representation in Table 8 of Appendix *(Reviewer K1pA)*. The experiments with higher-order transition functions, the comparison with MoLo, HySRM and HCL and the comparison between separate training and joint training are added in Table 12, Tables 6 and 7 and Table 13 of Appendix, respectively *(Reviewer K1pA)*;

We have corrected all typos mentioned in comments and any others we have found. *For optimal viewing of the paper, we recommend using Adobe Acrobat Reader. Please note that the original PDF files for Figures 1, 7, 8, and 9 are available in the supplementary materials, providing additional illustration and support for our assumptions.* We hope you may find our response satisfactory. Please let us know if you have any further feedback.

---

> ### Author Response · Authors · 2023-11-20
> **[Last few days for us to respond] Could you please go through and further comment on our response?**
>
> Dear Reviewers and Chairs,
>
> We express our gratitude for your efforts and comments. *We have provided detailed, point-to-point responses to each of your questions and comments and have uploaded a revised version of our paper*, with changes clearly marked in blue for your convenience.
>
> We hope that our revisions and responses adequately address your concerns. We are also fully prepared to respond to any additional queries you may have.
>
> As the deadline for discussion is nearing, we would be very grateful for your prompt further feedback.
>
> The authors of 342

---

> ### Comment · Area_Chair_c9pC · 2023-11-22
> **Discussion**
>
> Hi Reviewers, can you check the authors' reply to see if authors' have addressed your concerns?

---

### Meta-Review · Area_Chair_c9pC · 2023-12-07

**Metareview:**

This work introduces a few-shot action recognition method utilizing temporal casual representations. The angle seems to be novel. However, all the four reviewers pointed out that the paper is not significant enough to be accepted at present, and three of them tend to reject this work.
The major concerns include the motivations are not clear enough, and the background is not well introduced. There are also some assumptions made in the work, yet the validity also needs to be better justified. The AC believes this work has merits. However, the current version is not ready to be accepted. Thus, AC highly encourages the authors to carefully address the reviewers' comments, and then submit it to another top-tier conference.

**Justification For Why Not Higher Score:**

The major concerns include the motivations are not clear enough, and the background is not well introduced. There are also some assumptions made in the work, yet the validity also needs to be better justified.

**Justification For Why Not Lower Score:**

N/A

---

### Decision · Program_Chairs · 2024-01-16

Reject